# Climate change modulates the stratospheric volcanic sulfate aerosol lifecycle and radiative forcing from tropical eruptions

Thomas J. Aubry[1,2 ✉], John Staunton-Sykes[3], Lauren R. Marshall[3], Jim Haywood[4,5], Nathan Luke Abraham[3,6] & Anja Schmidt[1,3]

Explosive volcanic eruptions affect climate, but how climate change affects the stratospheric volcanic sulfate aerosol lifecycle and radiative forcing remains unexplored. We combine an eruptive column model with an aerosol-climate model to show that the stratospheric aerosol optical depth perturbation from frequent moderate-magnitude tropical eruptions (e.g. Nabro 2011) will be reduced by 75% in a high-end warming scenario compared to today, a consequence of future tropopause height rise and unchanged eruptive column height. In contrast, global-mean radiative forcing, stratospheric warming and surface cooling from infrequent large-magnitude tropical eruptions (e.g. Mt. Pinatubo 1991) will be exacerbated by 30%, 52 and 15% in the future, respectively. These changes are driven by an aerosol size decrease, mainly caused by the acceleration of the Brewer-Dobson circulation, and an increase in eruptive column height. Quantifying changes in both eruptive column dynamics and aerosol lifecycle is therefore key to assessing the climate response to future eruptions.

[1] Department of Geography, University of Cambridge, Cambridge, UK. [2] Sidney Sussex College, Cambridge, UK. [3] Department of Chemistry, University of Cambridge, Cambridge, UK. [4] College of Engineering Mathematics and Physical Sciences, University of Exeter, Exeter, UK. [5] Met Office Hadley Centre, Exeter, UK. [6] National Centre for Atmospheric Science, Leeds, UK. ✉email: ta460@cam.ac.uk

Explosive volcanic eruptions injecting sulfur gases into the stratosphere are one of the most important drivers of climate variability[1–3]. The sulfate aerosols produced by these eruptions reside for 1–3 years in the stratosphere where they scatter sunlight, resulting in a net negative radiative forcing at the top-of-the-atmosphere (TOA) and cooling at the surface[4,5]. Large-magnitude volcanic eruptions injecting on the order of 10 Tg of sulfur dioxide ($SO_2$) or more into the stratosphere and with a volcanic explosivity index[6] (VEI) greater than five are relatively rare events, with a return frequency on the order of decades[2,6,7]. They can, however, have a profound impact on climate, such as the Mount Tambora 1815 eruption, which was followed by the "year without a summer[8,9], or the Mount Pinatubo 1991 eruption, which resulted in 0.4–0.5 °C of global-mean lower-tropospheric cooling for over a year[10]. Large-magnitude eruptions may also affect key modes of climate variability such as the North Atlantic Oscillation[11], tropical monsoons[12,13], and the Atlantic meridional overturning circulation[14], and could affect the El Niño Southern Oscillation although the sign and magnitude of the response are debated[15,16]. Even relatively moderate-magnitude volcanic eruptions (VEI 3–5, injecting on the order of 1 Tg $SO_2$ or less in the upper troposphere-lower stratosphere, and with a return frequency on the order of a year[7,17], e.g., Nabro 2011) can affect Earth's radiation balance and temperatures. A cluster of such eruptions during 2005–2015 had a discernable cooling effect on lower tropospheric and sea surface temperatures[3,17,18].

Understanding the climatic effects of explosive eruptions has been a research focus for decades[19,20]. In contrast, understanding the impact of climate change on volcanic eruptions and potential feedback loops has received less attention, but is gaining momentum in the context of rapid global climate change driven primarily by anthropogenic greenhouse gas emissions. The processes and feedbacks by which climate change may affect explosive volcanic eruptions injecting sulfur into the stratosphere can be broadly classified into three categories:

1. The first category—the only one that has been subject to significant research efforts—relates to how changing climatic conditions affect the spatial, temporal, and magnitude distribution of explosive eruptions. In particular, there is substantial evidence that eruption frequency and magnitude increase following the deglaciation of ice-covered volcanoes[21]. However, the time-lag between ice retreat and the volcanic response has been constrained to 500–2000 years[22,23]. Consequently, such feedbacks are not relevant on the timescale of climate projections, including long-term projections (2100 to 2300)[24]. Other mechanisms have been proposed to affect eruption spatial and temporal distribution, such as changes in the hydrological cycle[25], but they remain debated[26].

2. The second category relates to how the background climate state affects the climate response to a volcanic eruption. This effect has long been discussed[27], but only recently has the impact of ongoing global climate change on the climatic response to future eruptions been investigated using climate models[28,29]. For example, as the Earth warms and ocean stratification increases, the surface cooling associated with a Mount Tambora (1815)-like eruption may not penetrate as deep into the lower ocean, resulting in enhanced cooling of the upper ocean and surface air[28]. On the contrary, one other study suggested that climate change will dampen the surface cooling associated with large tropical eruptions[29]. The Model Intercomparison Project on the climatic response to Volcanic forcing (VolMIP) includes one experiment testing how the climatic response to a large tropical eruption would differ in a preindustrial vs future climate[30]. However, one common limitation of the experimental designs of these studies[28–30] is

that they prescribe the perturbations in atmospheric optical properties induced by volcanic sulfate aerosols using constraints derived under present-day or preindustrial climate conditions which may not be valid under future climate conditions.

3. The third category relates to processes that directly govern the volcanic sulfate aerosol cycle and, in turn, perturbations in atmospheric optical properties and radiative forcing. A handful of studies have explored the impact of climate change on tropospheric volcanic sulfate aerosols[31,32] and how the climatic impacts of a large volcanic eruption would be modulated by stratospheric geoengineering via sulfur injections[33]. Only two studies have investigated how the stratospheric volcanic sulfate aerosol lifecycle would change in a warmer, non-geoengineered climate[34,35]. They show that the altitude at which volcanic aerosols and gas are injected is sensitive to climate change but do not quantify the subsequent impacts on radiative forcing and surface temperatures. The impact of climate change on all other processes directly governing stratospheric volcanic sulfate aerosol properties, including aerosol transport and microphysics[5,36], remains entirely unexplored.

The development of interactive stratospheric aerosol modeling capabilities—enabling the volcanic sulfate aerosol life cycle and radiative effects to be explicitly simulated—has been a major advance in a volcano–climate research in the last decade[36,37]. Furthermore, progress has also been made in the development and evaluation of eruptive column models (sometimes also termed "volcanic plume models")[38,39].

In this study, we utilize these developments and new capabilities to investigate how climate change will affect the stratospheric volcanic sulfate aerosol life cycle and radiative forcing. We use a global climate model with an interactive stratospheric aerosol module (UKESM1[40–42], used herein an atmosphere-only configuration) in combination with a one-dimensional (1D) eruptive column model[43] to analyze how both a moderate-magnitude tropical eruption and large-magnitude tropical eruption would affect climate in a present-day (1990–2000) and high-end future climate scenario (2090–2100, SSP5-8.5 scenario of the Scenario Model Intercomparison Project[24]). Figure 1, Table 1, and the Methods section detail our experimental design and some of the key characteristics of the two climate scenarios used, between which global-mean air surface temperature differs by 6.6 °C. We show that the global-mean stratospheric aerosol optical depth (SAOD) anomaly associated with the moderate-magnitude eruption decreases by a factor of 4 in the future. In contrast, for the large-magnitude tropical eruption, we find a larger SAOD and radiative forcing anomaly in the future climate, and, in turn, an amplification of post-eruption stratospheric warming response as well as tropospheric and surface cooling responses.

## Results

**Changes in $SO_2$ injection height**. In this subsection, using the eruptive column model, we show that the $SO_2$ injection height of a large-magnitude tropical eruption will increase by around 1.5 km in a warmer climate. The $SO_2$ injection height of a moderate-magnitude tropical eruption will remain unchanged but as a result of an increase in tropopause height, the plume spreads 2 km below the tropopause, reducing the amount of $SO_2$ injected directly into the stratosphere.

In UKESM1, volcanic eruptions are initialized by emitting a mass of $SO_2$ at a given latitude, longitude, and altitude (Fig. 1). Here we use a latitude and longitude corresponding to those of Mount Pinatubo (15.1°N,120.4°E), and $SO_2$ masses of 1 Tg of $SO_2$ and 10 Tg of $SO_2$ for our moderate-magnitude and large-magnitude

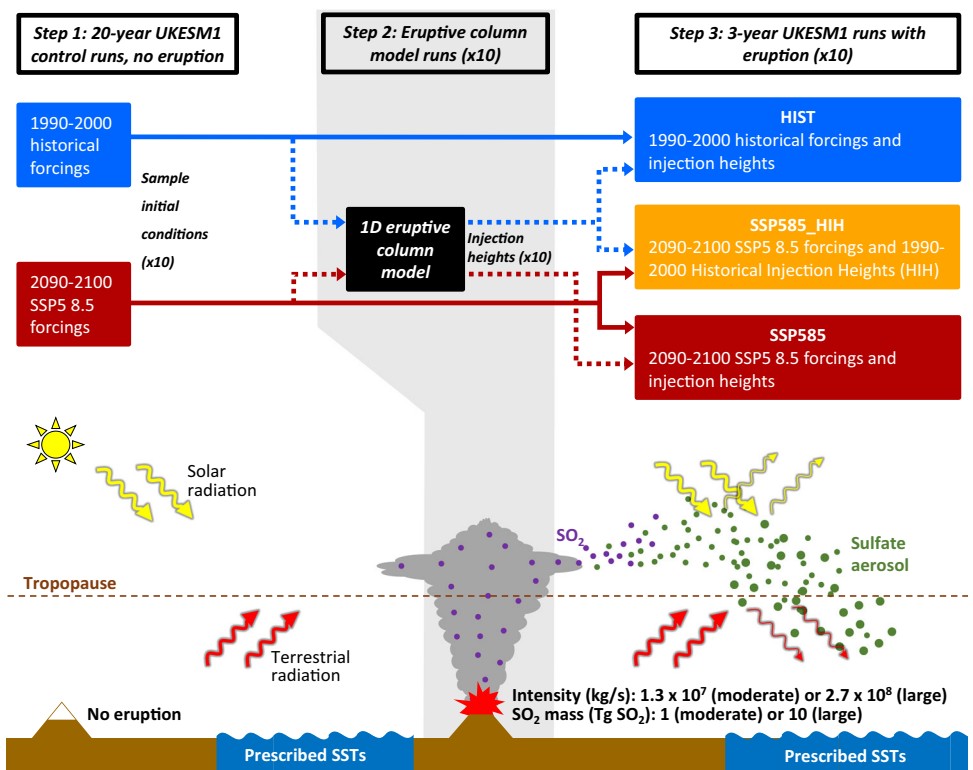

**Fig. 1 Overview of our combined eruptive column-aerosol-climate modeling framework and experimental design to produce the three scenarios analysed in this study.** Table 1 provides the $SO_2$ mass and eruption intensity used for each eruption scenario as well as key figures characterizing the HIST and SSP585 climate conditions. Further details on the models and design are provided in the Methods section. SST refers to sea-surface temperature and SSP5 8.5 is the high-end future climate scenario in projections of the UKESM1 Earth System Model.

eruption cases, respectively (Fig. 1 and Table 1). Previous studies using interactive stratospheric aerosol models to investigate the climatic impacts of volcanic eruptions directly prescribe the injection altitude[17,36,37]. In order to account for the effect of climate change on injection height[34,35], we first use an eruptive column model[43] and prescribe the eruption intensity (i.e., the mass eruption rate of ash and gas ejected through the vent) as $1.3 \times 10^7$ kg/s and $2.7 \times 10^8$ kg/s for our moderate and large-magnitude eruption cases, respectively. We use the intensity and the meteorological profiles (e.g., temperature or wind speed) simulated by UKESM1 on the eruption day at the eruption location as inputs to the eruptive column model (Fig. 1, Supplementary Figs. 1, 2, and Methods). Figure 2 shows key initial conditions, namely the distribution of $SO_2$ injection height and quasi-biennial oscillation (QBO) phase, for each of our four ensembles of simulations, corresponding to the moderate and large-magnitude eruption cases, and the historical 1990–2000 (HIST) and SSP5-8.5 2090–2100 (SSP585) climate scenarios. For the moderate-magnitude eruption and for both climate scenarios, $SO_2$ injection heights mostly range between 16 and 17 km (blue and red triangles in Fig. 2), in the vicinity of the HIST thermal tropopause (blue dashed line in Fig. 2), as observed for many recent moderate-magnitude tropical eruptions (e.g., Merapi 2010, Nabro 2011, Taal 2020). The lack of change in $SO_2$ injection height between the two climate states results in $SO_2$ being injected around 2 km below the tropical tropopause, the height of which increases by 1.5 km in the SSP585 scenario (red dashed line in Fig. 2). The tropical troposphere is more stratified and the upper troposphere and stratosphere less stratified in the SSP585 climate compared to HIST (Supplementary Fig. 1b). These effects compensate for eruptive columns reaching the vicinity of the tropopause resulting in similar average injection height for the SSP585 and HIST climate (Supplementary Fig. 2 and Aubry

et al.[34,35]). However, injection height is more variable for the SSP585 climate. In particular, two simulations are initialized with injection heights of 14–15 km (Fig. 2), much lower than the 16–17 km heights for other SSP585 simulations, which is a consequence of stronger than average tropospheric wind speed confining the eruptive column to the region of the troposphere where stratification increases (Supplementary Fig. 2b), which further decreases the column height[34,35]. Windy meteorological conditions also result in smaller injection heights in the HIST scenario, but this decrease is not further amplified by a weaker stratification. For the large-magnitude eruption case, $SO_2$ injection heights are close to 21 km in the HIST climate and 22.5 km for the SSP585 climate (blue and red circles in Fig. 2). This significant increase in injection height is a consequence of the decreasing stratospheric stratification in the SSP585 scenario (Supplementary Figs. 1, 2 and Aubry et al.[34,35]).

The injection heights determined using the eruptive column model (Fig. 2) are then implemented as the center heights at which $SO_2$ is injected in ensembles of ten simulations with UKESM1 (atmosphere-only configuration) for each pair of eruption cases (moderate and large magnitude) and climate scenario (HIST and SSP585) (Fig. 1, Table 1, and Methods). In addition, in order to disentangle the effects of changes in injection height from changes in the aerosol life cycle, we ran one additional set of ensembles for each eruption where UKESM1 is run with 2090–2100 SSP5-8.5 conditions, but HIST injection heights labeled SSP585_HIH (Historical Injection Height).

**Impact of climate change on the radiative forcing exerted by a moderate-magnitude tropical eruption.** Here we show that the peak monthly global-mean SAOD (at 550 nm) anomaly associated with a moderate-magnitude tropical eruption decreases by

**Table 1 Overview of the experiments conducted.**

| Experiment name | HIST, moderate-magnitude | HIST, large-magnitude | SSP585_HIH, moderate-magnitude | SSP585_HIH, large-magnitude | SSP585, moderate-magnitude | SSP585, large-magnitude |
|---|---|---|---|---|---|---|
| Mass of $SO_2$ (Tg) | 1 | 10 | 1 | 10 | 1 | 10 |
| Intensity (kg/s) | $1.3 \times 10^7$ | $2.7 \times 10^8$ | $1.3 \times 10^7$ | $2.7 \times 10^8$ | $1.3 \times 10^7$ | $2.7 \times 10^8$ |
| Eruption Location | Mount Pinatubo location (15.1°N,120.4°E) | | | | | |
| Ensemble size | 10 | | | | | |
| Atmospheric conditions used to calculate $SO_2$ injection height | Historical, 1990–2000 | | | | | |
| Climate scenario used to run UKESM1-AMIP | Historical, 1990–2000 | | SSP5 8.5, 2090–2100 | | SSP5 8.5, 2090–2100 | |
| Control run global-mean temperature (°C): sea surface \| surface air \| 50 hPa | 17.8 \| 13.7 \| −62.2 | | 22.5 \| 20.3 \| −63.5 | | | |
| Control run $CO_2$ concentration (ppm) | 360 | | 1060 | | | |
| Control run $CH_4$ concentration (ppb) | 700 | | 1080 | | | |
| Control run stratospheric Cl concentration (ppb) | 3.1 | | 1.1 | | | |
| Control run stratospheric Br concentration (ppb) | 0.02 | | 0.01 | | | |
| Control run $O_3$ concentration (DU) | 312 | | 343 | | | |
| Control run background global-mean aerosol optical depth at 550 nm: tropospheric \| stratospheric | 0.1188 \| 0.0035 | | 0.1159 \| 0.0022 | | | |

The bottom seven rows provide key metrics characterizing the forcings and climate state of the 20-year long control runs for the historical 1990–2000 and SSP5 8.5 2090–2100 climate scenarios. Unless specified, concentrations of chemical species are for the whole atmosphere. Control run values of aerosol optical depth are in the absence of any volcanic eruption.

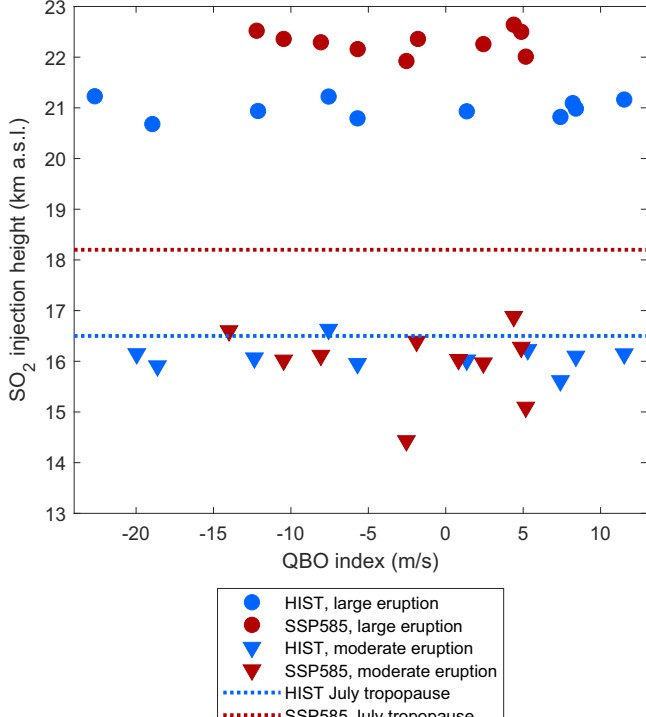

**Fig. 2 Initial conditions in the UK Earth System Model (UKESM) simulations conducted.** $SO_2$ injection height (km) as a function of the quasi-biennial oscillation (QBO) index (m s$^{-1}$) for the HIST (blue, historical climate) and SSP585 (red, high-end future climate) experiments. Dashed lines show the July-mean tropopause height at the eruption location (15.1° N,120.4°E) for the two climates. The QBO index was defined as the average zonal wind at 30 hPa between 15°S and 15°N. The injection height shown is the center height of the injection (see Methods section).

a factor of 4 in a future warmer climate as a consequence of reduced stratospheric $SO_2$ injections. The TOA and surface radiative forcings also decrease although their changes are small relative to natural variability for a single moderate-magnitude eruption.

Figure 3 shows the evolution of monthly global-mean SAOD (at 550 nm, Fig. 3a), TOA net (shortwave plus longwave) radiative forcing (Fig. 3c), and surface net radiative forcing (Fig. 3e) in response to the moderate-magnitude eruption, as well as time-averaged values of these three parameters for post-eruption year 1 and over years 1–3 (Fig. 3g). The response in the SSP585 and SSP585_HIH scenarios are very similar because of similar average $SO_2$ injection heights. For the HIST scenario (blue lines in Fig. 3), the global-mean SAOD (at 550 nm) anomaly peaks at 0.007–0.01, similar to observations for the 2011 eruption of Nabro[44]. Perturbations in the radiative forcing are challenging to detect for a single moderate-magnitude eruption because of the relatively small $SO_2$ mass injected (Fig. 3c, e), but observational and modeling studies have demonstrated that the combined effect of such frequent eruptions affects radiative forcing[17] and climate[3]. In the SSP585 scenario (red lines in Fig. 3), the peak global-mean SAOD is reduced by a factor of 4 compared to HIST. The time series of radiative forcing exhibits a large variability as for HIST, but the ensemble mean 3-year integrated TOA radiative forcing (Fig. 3g) is smaller by a factor of 3 in the SSP585 scenario compared to the HIST scenario (change significantly at the 80% significance level) and is even positive for the SSP585_HIH scenario (change significantly at the 95% confidence level). Changes in the SAOD anomaly are explained by the increase in

tropopause height combined with the unchanged $SO_2$ injection height (Fig. 2), resulting in a much smaller fraction of the total sulfur mass reaching the stratosphere. Figure 4 shows the ensemble-mean evolution of the sulfur (total in $SO_2$ and $H_2SO_4$) mass mixing ratio anomaly for the HIST (Fig. 4a–d) and SSP585 (Fig. 4e–h) scenarios, for the moderate-magnitude eruption. A large portion of the sulfur is initially injected slightly below the tropopause (black dashed line in Fig. 4) in the HIST scenario, but the self-lofting of the sulfate aerosol cloud due to stratospheric heating resulting from the absorption of radiation at infrared and near-infrared wavelengths results in most of the sulfur (60%) being transported into the stratosphere. In contrast, in the SSP585 scenario, most of the sulfur cloud does not rise high enough to compensate for the 2-km difference between the $SO_2$ injection height and the tropopause height, resulting in a smaller fraction of the total injected sulfur (25%) being in the stratosphere 1 month after the eruption.

**Impact of climate change on the radiative forcing exerted by a large-magnitude tropical eruption.** Figure 3 shows the evolution of global-mean SAOD (Fig. 3b), TOA radiative forcing (Fig. 3d), and surface radiative forcing (Fig. 3f) in response to the large-magnitude eruption. In both the SSP585 (red lines in Fig. 3b) and SSP585_HIH (orange lines) scenario, the peak global-mean SAOD increases by around 10% relative to HIST (blue lines), the magnitude of the peak global-mean TOA radiative forcing increases by around 30% and that of peak global-mean surface radiative forcing increases by 18% (SSP585) to 35% (SSP585_HIH). However, global-mean SAOD decreases faster over time, in particular in the SSP585_HIH scenario. This is apparent in the time-integrated global-mean SAOD values (Fig. 3h) with a significant increase in SSP585 compared to HIST for the first post-eruption year (shaded bars), but not for the first 3 post-eruption years taken altogether (opaque bars). These changes are smaller and less significant when we do not account for changes in $SO_2$ injection height (SSP585_HIH). Similar changes are evident for the TOA radiative forcing, with an increase of 21% for the first post-eruption year in both the SSP585 and SSP585_HIH scenarios. Finally, the time-integrated global-mean surface radiative forcing shows statistically significant increases (+16–27%) for both year 1 and year 1–3, regardless of whether we account for changes in $SO_2$ injection height.

**Impact of climate change on volcanic sulfate aerosol life cycle for the large-magnitude tropical eruption.** In this subsection, we show that the increase in radiative forcing for the large-magnitude eruption is driven by the net effect of two competing factors: a decrease of the sulfate aerosol size and of the sulfate aerosol lifetime in the stratosphere. Both changes are mostly driven by the acceleration of the Brewer–Dobson circulation and modulated by the increase of the $SO_2$ injection height and aerosol nucleation rate.

Regardless of whether we account for changes in $SO_2$ injection heights, the volcanic $SO_2$ e-folding time decreases by 23% from 40 days in the HIST scenario to 31 days in the SSP585 scenario (Fig. 5a, see methods for the definition of e-folding times). This change is likely related to higher OH concentrations in the future (Supplementary Fig. 4c, d, themselves caused by increasing stratospheric water vapor concentrations, Supplementary Fig. 4e, f), and consequently faster conversion of $SO_2$ into sulfuric acid ($H_2SO_4$) via gas-phase oxidation. The global $H_2SO_4$ burden (Fig. 5b) is smaller and decays faster in the SSP585 and SSP585_HIH scenarios compared to HIST, with changes in SSP585_HIH being more pronounced. The global total ($SO_2$ + $H_2SO_4$) S burden e-folding time $\tau_S$ (see methods), decreases from 15.8 months in the HIST scenario to 14.2 months in the

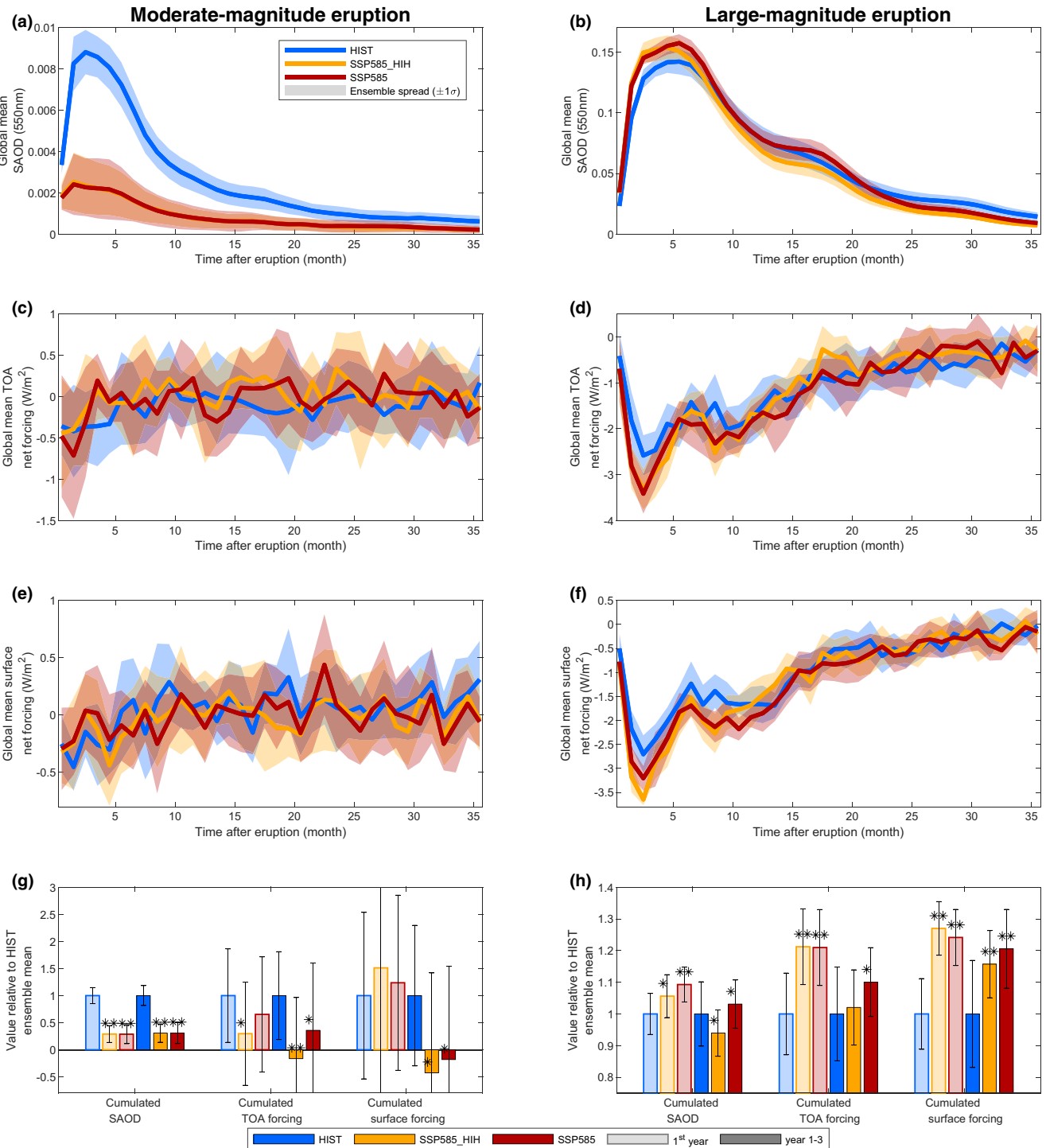

**Fig. 3 Stratospheric aerosol optical depth (SAOD) and radiative forcing in our simulations.** Time series of global-mean SAOD anomaly at 550 nm (**a**, **b**), global-mean net top-of-the-atmosphere (TOA) radiative flux anomaly (**c**, **d**, in W m$^{-2}$), and global-mean net surface radiative flux anomaly (**e**, **f**, in W m$^{-2}$) for the three scenarios. Thick lines show the ensemble mean and shading shows the spread as one standard deviation across ensemble members. The bottom bar graphs (**g**, **h**) show time-integrated values of the above time series for the first post-eruption year and the first 3 post-eruption years, normalized by the ensemble mean for the HIST scenario. Error bars for the bottom bar graph show the ensemble spread; a single (double) star indicates that the change relative to the HIST experiment is significant at the 80% (95%) level. Left plots (**a**, **c**, **e**, **g**) show results for the moderate-magnitude eruption (1 Tg), right plots (**b**, **d**, **f**, **h**) show results for the large-magnitude eruption (10 Tg). The HIST scenario represents the historical climate and the SSP585 scenario is an upper-end future climate scenario. SSP585_HIH is as SSP585 but uses SO$_2$ injection heights consistent with the HIST scenario.

SSP585 scenario and 12.2 months in SSP585_HIH (Fig. 5c, all changes significant). These changes in e-fold explain the faster decay of SAOD and radiative forcing (Fig. 3). However, they cannot explain the increases in global-mean peak SAOD and

TOA radiative forcing. Figure 5d shows the global-mean stratospheric aerosol effective radius time series, with a peak value of 0.41 μm for HIST, which is consistent with observations following the 1991 eruption of Mount Pinatubo[44,45]. Over the first 2

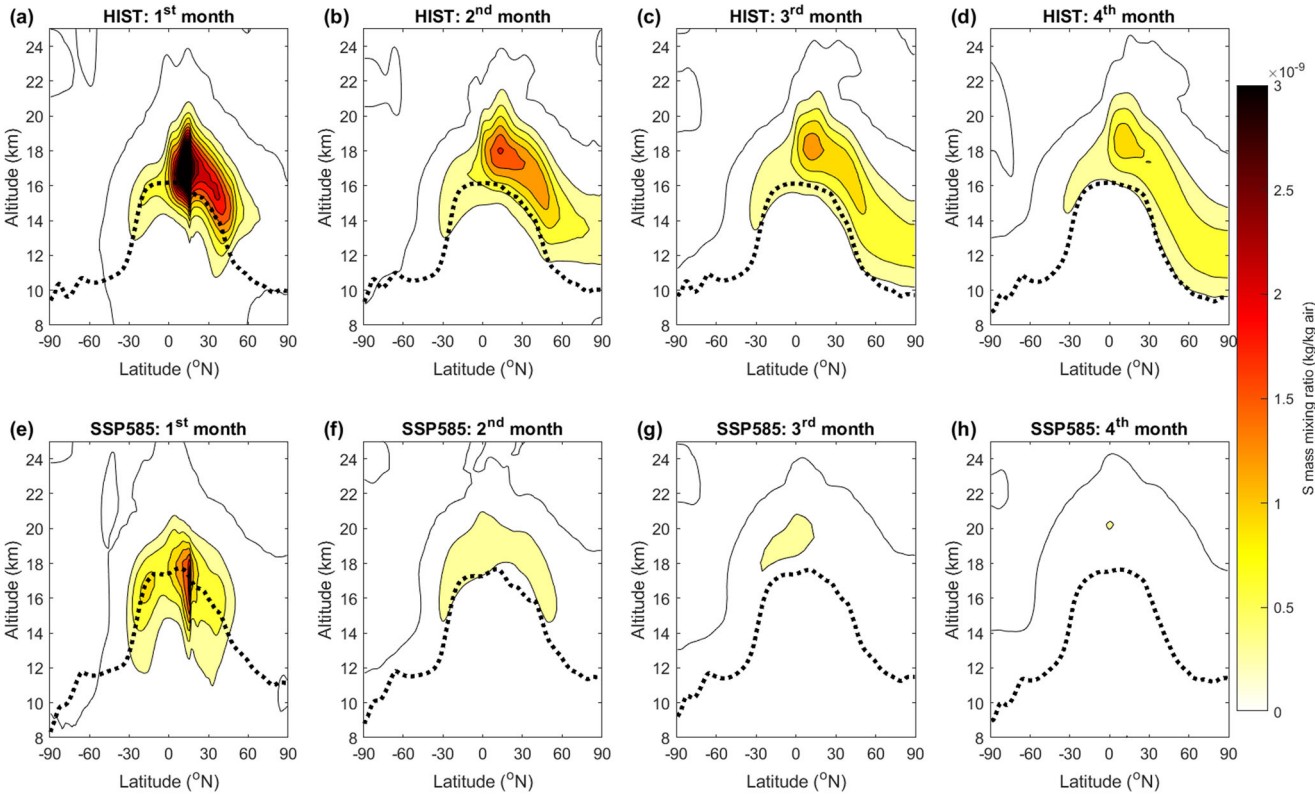

**Fig. 4 Initial sulfur distribution for the moderate-magnitude eruption.** Zonal mean mass mixing ratio (kg/kg of air) of S (in $SO_2$ and $H_2SO_4$) for each of the first 4 post-eruption months for the HIST (**a–d**, historical climate) and SSP585 (**e–h**, high-end future climate) experiments. Results shown are for the moderate-magnitude tropical eruption and are averaged across all ensemble members. The black dotted line shows the thermal tropopause height, which is up to 1.75 km higher in SSP585 compared to HIST. Significance levels are not displayed for clarity, but differences between the two scenarios are significant at the 95% level for the vast majority of the HIST region with positive mass mixing ratio anomalies.

post-eruption years, the effective radius is smaller by up to 11% in the SSP585 scenario compared to HIST and smaller by up to 18% in the SSP585_HIH scenario compared to SSP585. These changes are most significant from the 8th month following the eruption and explain well the evolution of simulated SAOD and radiative forcing, with a smaller effective radius resulting in more efficient scattering per unit mass at short (solar) wavelengths and thus greater SAOD (at 550 nm) and radiative forcing[36], in particular during the first eruption year. Later on, the smaller effective radius cannot compensate for the faster removal of aerosol in SSP585 and SSP585_HIH scenarios, explaining the smaller SAOD and forcing values. Smaller-sized aerosols are removed more slowly from the stratosphere via gravitational settling[36,46]. This effect holds in UKESM1 but in our simulations, it is dominated by the accelerated removal driven by the faster Brewer–Dobson circulation, which we further discuss below.

While changes in S burden e-folding time and aerosol effective radius explain well the temporal evolution of the radiative forcing (Fig. 3), we have yet to discuss how these two key aerosol properties are affected by climate change. Figure 6a–f shows the time-latitude evolution of the $H_2SO_4$ column burden for all scenarios (see also Supplementary Fig. 5 for the vertical evolution). The sulfuric acid aerosols spread faster to higher latitudes in the SSP585_HIH scenario compared to the historical scenario (Fig. 6d), which in turn decreases the S burden e-folding time as sulfate aerosols sediment from the stratosphere into the troposphere predominantly at mid-high latitudes[4,5]. This faster spreading is the consequence of the acceleration of the Brewer–Dobson circulation under climate change[47] (Supplementary Fig. 4g, h). When we account for the changes in plume

dynamics resulting in higher injection height (SSP585, Fig. 6c, e, f), this effect is partially compensated by the decreasing speed of the Brewer–Dobson circulation with increasing altitude[47].

In the modal aerosol scheme implemented in UKESM1[48], volcanic sulfate aerosols are most efficient at scattering shortwave radiation when they are in the accumulation mode with mean radii between 0.05 and 0.5 μm. As these sulfate aerosol particles grow, they are transferred to the coarse mode (with mean radii exceeding 0.5 μm) where their scattering efficiency at solar wavelengths strongly decreases and resulting in particles that settle faster[48]. Figure 7a shows that the flux of sulfate aerosol out of the accumulation mode into the coarse mode is significantly larger in the HIST scenario compared to SSP585, and significantly larger in the SSP585 scenario compared to SSP585_HIH, in particular between the 8th to 20th post-eruption month. This in turn leads to a larger effective radius in HIST compared to SSP585, and SSP585 compared to SSP585_HIH (Figs. 4 and 5).

Two main hypotheses can explain the stark differences in sulfate aerosol growth between the HIST and SSP585 scenarios. The first one is related to large-scale atmospheric transport: aerosol particles that are confined for longer in the tropical pipe grow larger, whereas outside the tropical pipe rapid spreading to high latitudes results in reduced condensational and coagulation growth, and thus, on average, smaller effective radii[49,50]. This is supported by the larger-sized aerosol particles developing in the tropical pipe in the historical scenario compared to SSP585_HIH, and in the SSP585 scenario (higher plume height and slower spreading) compared to SSP585_HIH (Fig. 6g–l). The transport hypothesis is further supported by Fig. 7b, which shows a scatter plot of the 3-year mean effective radius as a function of the tropical S burden

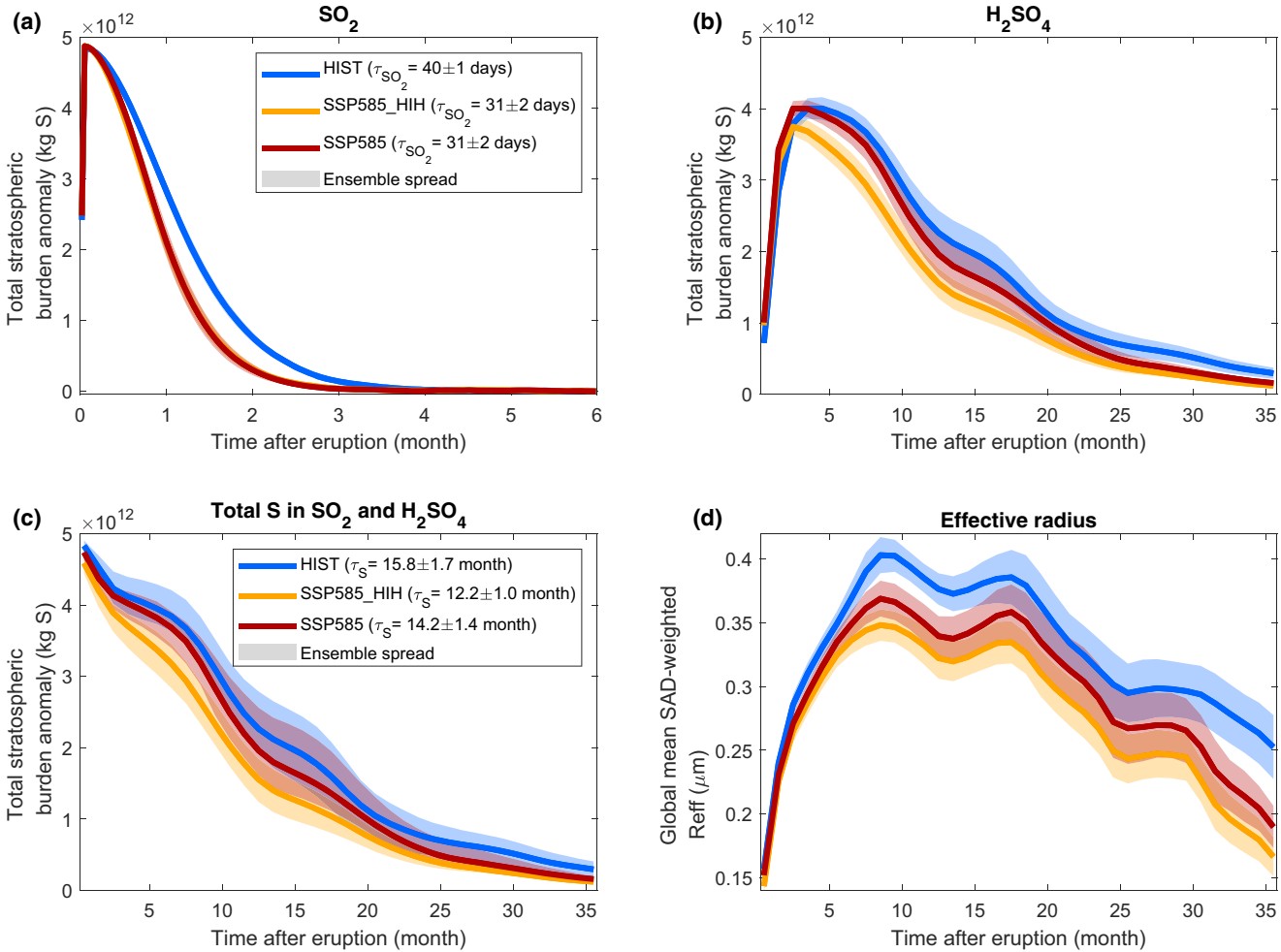

**Fig. 5 Temporal evolution of the global-mean stratospheric S budget (Tg S) and the global-mean sulfate aerosol effective radius (Reff in μm) for the large-magnitude eruption.** Time series of total stratospheric burden anomalies of $SO_2$ (**a**), $H_2SO_4$ (**b**), and S (**c**). Panel **d** shows the stratospheric global-mean surface area density-weighted effective radius time series. The different colors correspond to the different scenarios. All panels are for the large-magnitude eruption. The $SO_2$ time series is a daily mean, all other ones are monthly means. See Methods section for definitions of the $SO_2$ and total S e-folding time provided. The HIST scenario represents the historical climate and the SSP585 scenario is an upper-end future climate scenario. SSP585_HIH is as SSP585 but uses $SO_2$ injection heights consistent with the HIST scenario.

e-folding time, with these two variables significantly correlated ($p$ value <0.0001) both across all simulations and across members of individual ensembles. The second hypothesis is related to aerosol microphysics. In the first 2 post-eruption months, the aerosol nucleation rate is two-three times greater in the SSP585 scenarios compared to the historical scenario (Fig. 7c). This results in more numerous but smaller aerosol particles which contributes to the smaller effective radius simulated under a warmer climate. Nucleation rates of sulfate aerosol increase with colder temperatures[36,48], which is consistent with colder stratospheric temperatures (Supplementary Fig. 4a, b) projected under SSP585 scenarios compared to HIST. Significant differences in effective radius between the SSP585_HIH and SSP585 scenarios can be explained by our transport hypothesis, but not by nucleation rate changes as they are not significantly different for these two scenarios (Fig. 7c). The high value of correlation coefficients between the tropical S burden e-folding time and the average effective radius (r > 0.91, Fig. 7b) also suggests that the residence time in the tropical pipe is the dominant driver of effective radius variability across these experiments with a fixed erupted $SO_2$ mass. Both hypotheses described above can explain the evolution of aerosol size in the tropics during the first 6 post-

eruption months (Fig. 6g–l). As volcanic sulfate aerosol particles grow by coagulation and condensation, aerosols are transported to higher latitudes, where the relatively larger-sized aerosol in the HIST (and to a lesser extent SSP585) scenario have a higher rate of transfer out of the accumulation mode into the coarse mode. As an example, Supplementary Fig. 6 shows the flux of aerosol into the coarse mode resulting from the coagulation of particles from the accumulation mode. The coagulation rate is much greater in the HIST scenario compared to the SSP585 scenario, and in the SSP585 scenario compared to the SSP585_HIH scenario. This causes the effective radius to be larger in HIST (Figs. 4 and 5) and, in turn, SAOD and radiative forcing to be smaller (Fig. 3b, d, f, h).

**Large-magnitude tropical eruption: changes in the climatic response.** In this subsection, we show that in response to the increased forcing for a large-magnitude eruption in a warmer climate, there is an amplification of the subsequent stratospheric warming, tropospheric cooling, and surface cooling. Changes in eruptive column dynamics and $SO_2$ injection height contribute to this amplified temperature response.

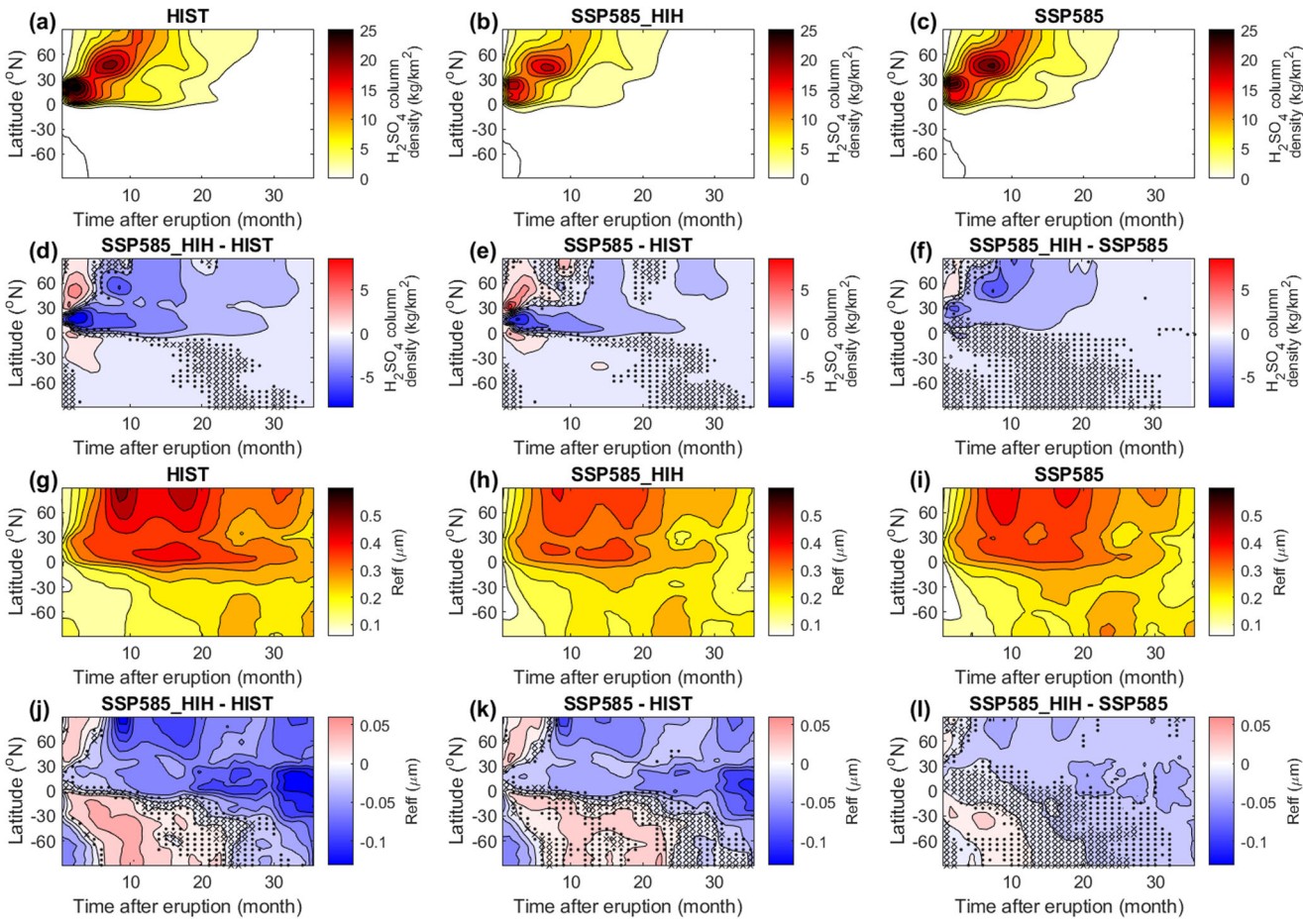

**Fig. 6 Time-latitude evolution of the H$_2$SO$_4$ column density and aerosol effective radius for the large-magnitude eruption. a–f** H$_2$SO$_4$ column density (kg m$^{-2}$). **g–l** Aerosol effective radius (μm). Individual panels either show values for one of the three scenarios or the pairwise difference between scenarios and are labeled accordingly. For panels showing the pairwise difference, dots highlight areas where changes are not significant at 95%, and crosses highlight areas where the difference is not significant at the 80% level. The HIST scenario represents the historical climate and the SSP585 scenario is an upper-end future climate scenario. SSP585_HIH is as SSP585 but uses SO$_2$ injection heights consistent with the HIST scenario.

One of the best-understood features of the climatic response to volcanic eruptions when globally averaged is a cooling of the surface and troposphere and a warming of the stratosphere. Figure 8a shows that the global-mean stratospheric warming is significantly stronger by 55% in both the SSP585 and SSP585_HIH scenarios during the first post-eruption year, with stratospheric temperature anomalies increasing by up to 2° (Fig. 8d–f). Accounting for changes in SO$_2$ injection height (SSP585) results in persistently higher stratospheric temperature anomalies during the second and third post-eruption year, so that the post-eruption stratospheric temperature anomaly remains significantly larger by 55% over 3 years in SSP585 compared to HIST (Fig. 8a). By contrast, in the SSP585_HIH scenario, the 3-year mean stratospheric warming is not significantly different from HIST because of the aerosol burden, SAOD and forcing decay faster due to the lower injection height. Figure 8b shows that similarly, mid-tropospheric cooling is amplified by around 80% in the SSP scenarios during the first post-eruption year and 30% (SSP585_HIH) to 50% (SSP585) over the first 3 post-eruption years, although this change is only significant at the 80% significance level (or insignificant for the 3-year average for SSP585_HIH). Figure 8d–f shows the spatial distribution of temperature anomalies for the first post-eruption year in more detail. In SSP585_HIH (Fig. 8e), the amplified stratospheric warming and tropospheric cooling is significant at the 95% level in only a small region (non-hatched areas, mostly between 45°N

and 60°N throughout the stratosphere). In contrast, in the SSP585 scenario accounting for changes in injection height (Fig. 8f), the change in temperature anomalies is significant throughout most of the tropics and northern hemisphere between 10 and 50 hPa, and in the tropical troposphere between 100 and 400 hPa. Finally, as we conduct atmosphere-only simulations with prescribed sea surface temperatures, we cannot directly diagnose the surface temperature response. To do so, we use the annual global-mean net TOA radiative forcing anomaly time series obtained from our UKESM1 atmosphere-only experiments together with the simple climate model FaIR[51,52] (see Methods) to estimate the global-mean surface temperature anomaly. Results are shown on Fig. 8c and suggest that averaged over the first 3 post-eruption year, the 0.2 °C global-mean surface cooling is amplified by 7% for SSP585_HIH (change not significant) and 15% for SSP585 (change significant at 95% level) compared to HIST. For both scenarios, the surface cooling during the first post-eruption year is amplified by 21% (significant at 95% level).

Beyond the surface temperature response, the simulated changes in sulfate aerosol life cycle and volcanic radiative forcing may affect the dynamical response of the upper atmosphere to volcanic eruptions. For example, a strengthening of the polar vortex following tropical eruptions might lead to a positive North Atlantic Oscillation phase[11,53]. Preliminary results (Supplementary Fig. 7) suggest that for the second post-eruption winter, there is a significant strengthening of the polar vortex in the HIST

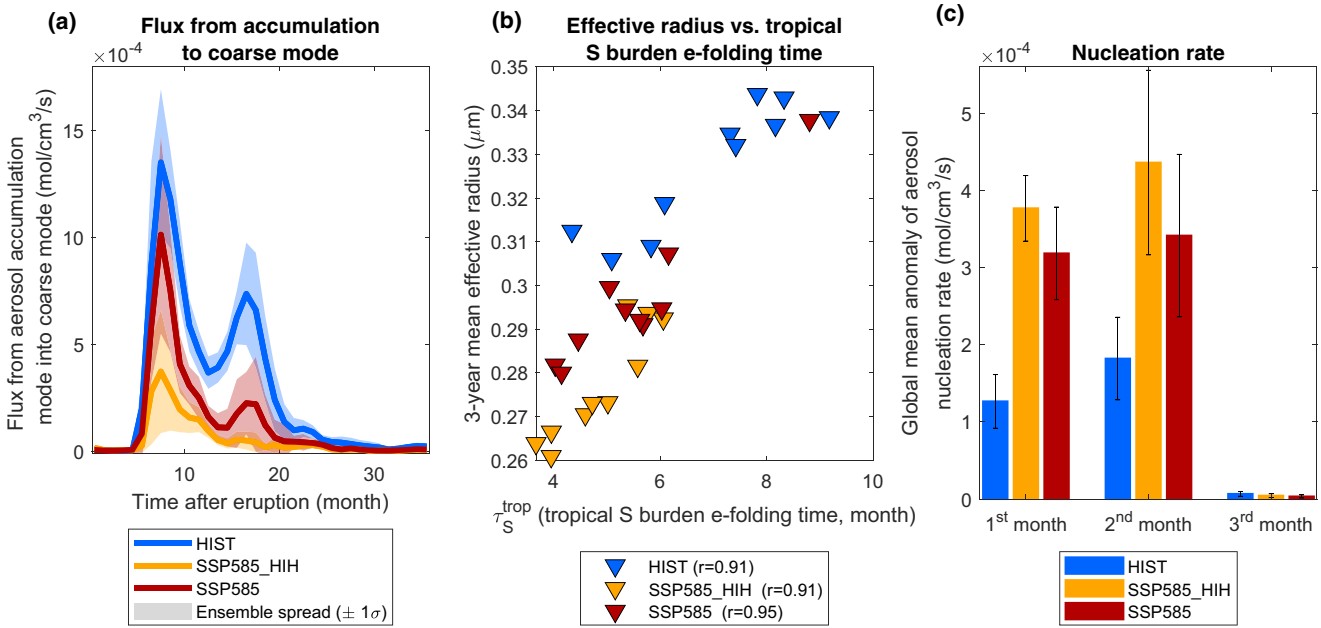

**Fig. 7 Key factors influencing the aerosol effective radius evolution for the large-magnitude eruption.** Left **a** time series of the global-mean flux from the aerosol accumulation mode into the aerosol coarse mode (mol cm$^{-3}$ s$^{-1}$). Center **b** 3-year global-mean aerosol effective radius (μm) as a function of the tropical S burden e-folding time (month). Right **c** Global-mean anomaly of binary homogeneous nucleation rate of sulfuric acid and water (mol cm$^{-3}$ s$^{-1}$) for the first 3 post-eruption months. Error bars show one standard deviation across ensemble members. The HIST scenario represents the historical climate and the SSP585 scenario is an upper-end future climate scenario. SSP585_HIH is as SSP585 but uses SO$_2$ injection heights consistent with the HIST scenario.

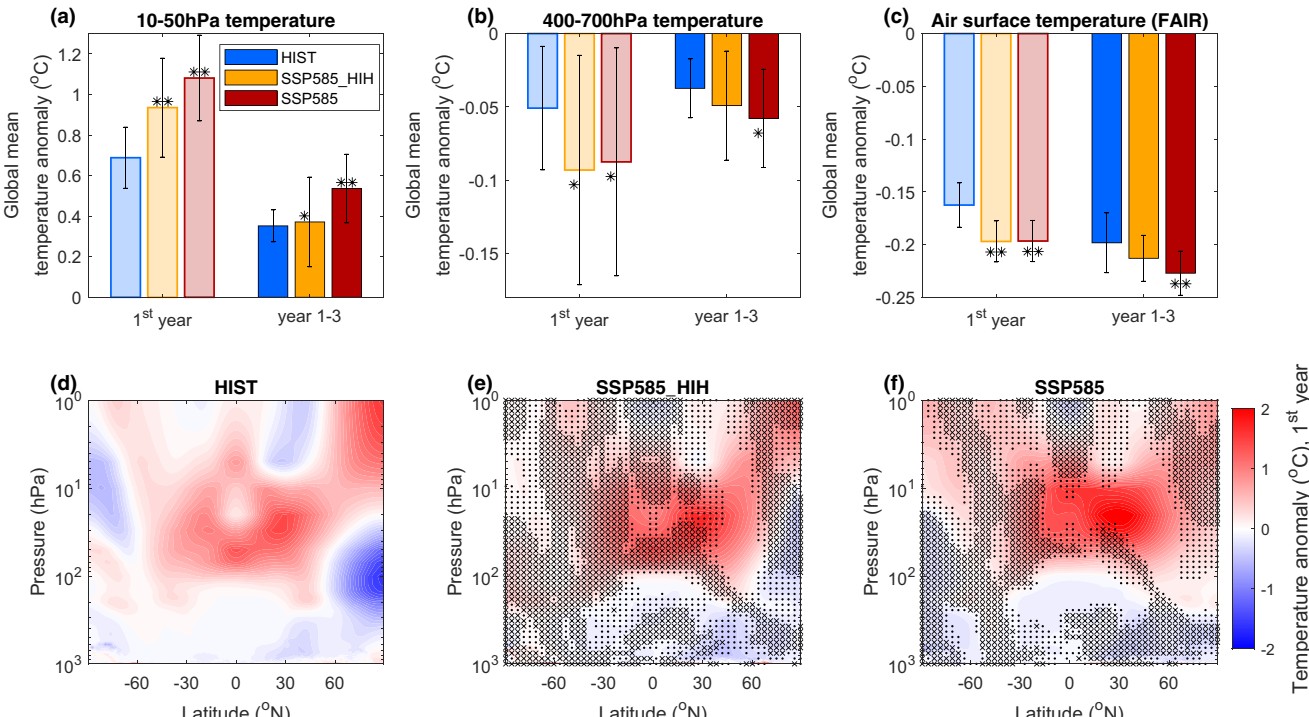

**Fig. 8 Temperature response to the large-magnitude tropical eruption.** Top row **a-c** Bar graphs showing the global-mean mid-stratospheric (10–50 hPa) temperature anomaly (**a**) the global-mean mid-tropospheric (400–700 hPa) temperature anomaly (**b**), and the global mean air surface temperature anomaly as estimated from the net TOA radiative forcing using the simple emission-based climate model FaIR[51, 52] (see Methods) (**c**) for the first post-eruption year and the year 1–3 average. Error bars show one standard deviation across the ensemble members; a single (double) star indicates that the change relative to the HIST experiment is significant at the 80% (95%) level. Bottom row **d-f** zonal mean temperature anomaly averaged over the first post-eruption year for the HIST (**d**), SSP585_HIH (**e**), and SSP585 (**f**) scenarios. Dots indicate areas, where changes are not significant at the 95% and crosses, indicate areas where the difference is not significant at the 80% level. The HIST scenario represents the historical climate and the SSP585 scenario is an upper-end future climate scenario. SSP585_HIH is as SSP585 but uses SO$_2$ injection heights consistent with the HIST scenario.

scenario but no response (SSP585) or a weakening (SSP585_HIH) of the polar vortex in the future scenarios. Stratospheric volcanic sulfate aerosols can also deplete ozone ($O_3$) by providing a surface upon which heterogeneous chemical reactions between anthropogenic chlorofluorocarbons (CFC) gases and $O_3$ can take place[4,5]. In our simulations, global-mean ozone depletion is three times stronger in the HIST climate (Supplementary Fig. 8) which is expected as consequence of the much lower background CFC concentrations in the SSP585 scenario compared to HIST[54].

Lastly, some of the mechanisms via which climate change affects the volcanic sulfate aerosol cycle and forcing are also affected by the volcanic aerosol themselves. For example, the Brewer–Dobson circulation is accelerated by both anthropogenic greenhouse gas emissions[47] and volcanic aerosol from tropical eruptions[55,56]. Our own experiments are consistent with an accelerated circulation following the large-magnitude eruption, with a decrease of stratospheric age of air in the Northern Hemisphere in all scenarios (Supplementary Fig. 9). Whereas the forcing and stratospheric warming is enhanced in the future climate, we find a more pronounced age of air anomalies in the HIST scenario. The volcanically forced acceleration of the Brewer–Dobson circulation, as well as the strengthening of the winter polar vortex, have been linked to enhanced wave propagation as a result of aerosol heating and the increase of the meridional stratospheric temperature gradient[57]. Our results are consistent with this picture as the HIST scenario is characterized by a slower transport of the aerosol cloud to high-latitudes (Fig. 6a–f) resulting in a stronger meridional stratospheric temperature gradient as aerosols reside in the tropics for a longer period of time (Fig. 9d–f). This in turn explains both a stronger acceleration of the Brewer–Dobson circulation (Supplementary Fig. 9) and strengthening of the polar vortex (Supplementary Fig. 7) compared to the SSP585 and SSP585_HIH scenarios. The eruption-induced change in the age of air (ca. $-0.25$ years between 20 and 30 km altitude, Supplementary Fig. 9) is relatively small compared to the difference between the two climate scenarios considered (ca. $-1$ year, Supplementary Fig. 4g, h).

## Discussion

Figure 9 summarizes the mechanisms via which climate change affects a moderate-magnitude and large-magnitude tropical eruption. In this section, we discuss the robustness of our results, their implications for understanding climate–volcano interactions, as well as some key future research directions.

Our simulations for the moderate-magnitude tropical eruption case suggest damping of the peak global-mean SAOD anomaly by a factor of 4 in a high-end future climate scenario (SSP585). Consequently, we project a decrease of the background tropical stratospheric aerosol layer, which is largely governed by sulfur injections by moderate-magnitude eruptions, and the radiative forcing it exerts on climate[17,18]. This projection is the result of three factors. First, it is driven by an increase of the tropical tropopause height in the future, which has been observed and is a consensual feature of climate model projections[58,59]. The early development of the volcanic cloud itself may lower the tropopause height, but any such effect driven by sulfur species is accounted for in our simulations, and both our simulations and observations[60] suggest a relatively small decrease of the tropopause height (on the order of 100 m) compared to the 1.5 km increase between our HIST and SSP585 scenarios. Second, the self-lofting caused by the absorption of radiation in the volcanic sulfur cloud[61] (Fig. 4) must be small enough to not compensate for the rise in tropopause height, an effect which may be dependent on the radiation scheme used in climate models. The

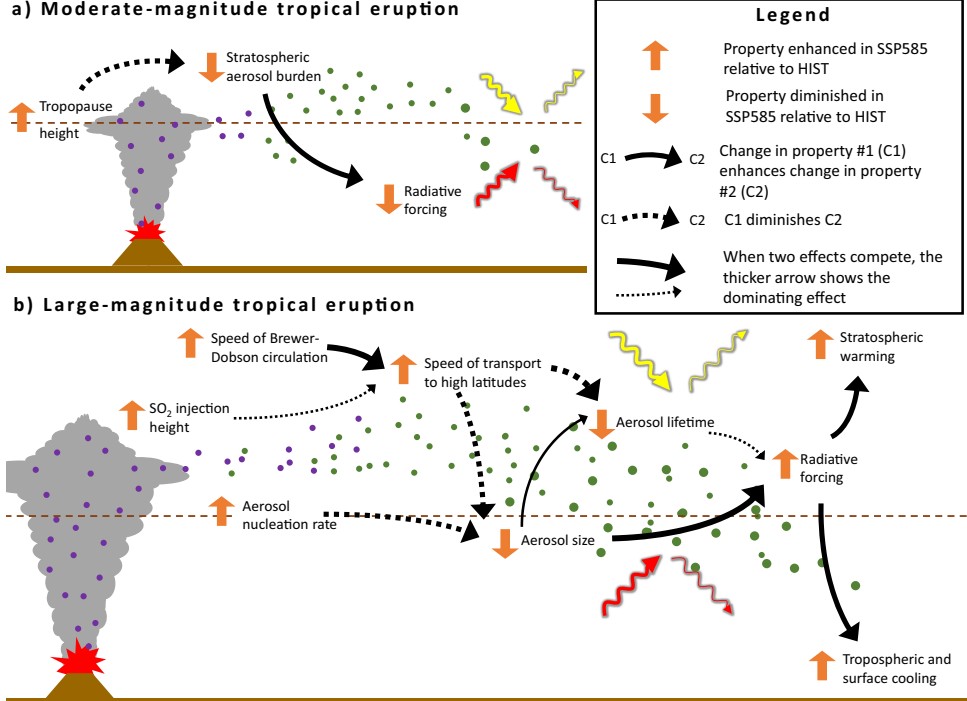

**Fig. 9 Schematic summarizing how climate change affects volcanic forcing.** Panel (**a**) refers to a moderate-magnitude tropical eruption and panel (**b**) refers to a large-magnitude tropical eruption. The symbols showing processes related to the volcanic sulfate aerosol cycle are identical to Fig. 1 (e.g., violet dots symbolize $SO_2$, green dots symbolize sulfate aerosols). Symbols indicating how various properties are affected by climate changes or affect each other are defined in the legend.

QBO phase may also modulate the sulfur cloud lofting[62]. Third, the eruptive column height of moderate-magnitude eruptions must be stable or decrease, and thus not compensate for the tropopause height increase. Eruptions of corresponding intensity are particularly sensitive to the strength of atmospheric wind and are impacted by uncertainties related to parameterization of turbulent entrainment of air into the plume (Supplementary Fig. 3a). More importantly, accounting for the impact of water phase change on the plume buoyancy flux may partly compensate the rise in tropopause height (Supplementary Fig. 3b) but this effect may be overestimated in 1D eruptive column models[39]. The role of water phase change in future plume height projections remains challenging to quantify as this process is unsatisfyingly represented in 1D eruptive column models[39] and is either not represented or not systematically evaluated in 3D models.

Our projections for the large-magnitude tropical eruption suggest a significant enhancement of the forcing (+10, +30, and +23 increase of the magnitude of peak global SAOD, TOA radiative forcing, and surface radiative forcing) and climatic impacts (52, 55, and 15% amplification of the global-mean stratospheric warming, tropospheric cooling, and land surface cooling averaged over the first 3 post-eruption years) in a high-end future climate scenario (SSP585). In our model simulations, the acceleration of the Brewer–Dobson circulation is a key driver of these changes (Fig. 9), resulting in smaller-sized aerosol particles and a decrease of the S burden e-folding time. The acceleration of the Brewer–Dobson circulation is a consensual feature of climate model projections[47], even though the exact magnitude of this acceleration is model-dependent. Furthermore, we also expect that a decrease in S e-folding time as a result of faster transport to high latitudes will also hold across all models with prognostic capabilities for stratospheric sulfate aerosols. However, the response of the aerosol effective radius to this acceleration may be model-dependent as effective radius remains one of the properties with the most discrepancies across interactive stratospheric aerosol models[30,46]. Accounting for changes in $SO_2$ injection height has a substantial impact on the effective radius and the total S burden e-folding time (Fig. 7), and it further amplifies the post-eruption temperature response (Fig. 8). For a tropical large-magnitude eruption, the projected increase in plume height is a robust feature with respect to both intra-model (Supplementary Fig. 3) and inter-model[35] uncertainties, even though the exact magnitude depends on the eruptive column model and climate model used[34,35]. In particular, uncertainty in turbulent entrainment parameterization (Supplementary Fig 3a) and water phase change (Supplementary Fig. 3b) do not affect the projected increase in plume height by ca. 1.5 km for large-magnitude tropical eruptions. Another factor that could affect the climate response to volcanic eruptions is tropospheric aerosol background state: using the previous generation (CMIP5) of model (HadGEM2-ES) and previous mid-high end emission scenario (Representative Concentration Pathway 6.0), Hopcroft et al. (2017)[29] showed that an increase in tropospheric aerosol between a preindustrial climate state and year 2045 results in an increased planetary albedo and reduced radiative forcing and surface cooling. Such a feedback mechanism is accounted for in our simulations, but we find a minor decrease (as opposed to the increase in ref. [29], or in the Community Earth System Model[63]) in tropospheric aerosol optical depth at 550 nm between our two chosen climate states (−2% from HIST to SSP585, Table 1). As such, we expect the change in tropospheric aerosol to play a negligible role in our results when compared to the processes governing the stratospheric aerosol life cycle.

There are two potential improvements in our combined plume-aerosol-climate modeling framework which may impact the results from both eruption scenarios explored. First, we only inject $SO_2$ to simulate a volcanic eruption in UKESM. Previous studies have shown that the co-injection of other volcanic products, including ash[64], halogens[65,66], and water[67], would affect the sulfate aerosol cycle including the $SO_2$ lifetime or the self-lofting of the aerosol cloud. Second, we used atmosphere-only simulations, so that feedbacks related to the ocean response are not included. The only study to have explored such feedbacks found that for a prescribed volcanic sulfate aerosol forcing corresponding to a Mount Tambora (1815)-like eruption, changes in ocean stratification amplify the surface cooling associated with the eruption in a future climate[28]. We expect that the same feedback would be at play in the fully-coupled version of UKESM1 and other models as a decrease in ocean stratification is a consensual prediction of global climate models under future climate change. Taken together, the combined enhancement of the surface temperature response driven by an increase in radiative forcing (+15%, this study, estimated using FaIR) and an increase in the surface temperature response to the forcing related to ocean feedbacks (+40%[28]) could lead to a 60% greater surface cooling for large-magnitude tropical eruptions. Experiments combining eruptive column modeling, interactive stratospheric aerosol modeling, and coupled ocean-atmosphere models are required to refine this back-of-the-envelope calculation. However, the large enhancement obtained suggests that climate–volcano feedbacks could modulate the decadal scale climate variability driven by explosive eruptions and future climate projections. Our work thus challenges the use of a constant volcanic forcing in future climate projections[24]. Furthermore, as we have demonstrated that the impact of climate change on volcanic forcing depends on the type of eruption considered (Figs. 3 and 9), our work does not clarify yet whether an increase or decrease in the long-term mean volcanic radiative forcing is expected in the future. A back-of-the-envelope calculation (see Methods) suggests that the long-term radiative forcing from eruptions injecting on the order of 10 Tg $SO_2$ or more would increase (in magnitude) from −0.29 W/m² under HIST conditions to −0.32 W/m² under SSP585 conditions, whereas that from eruptions injecting on the order of 1 Tg $SO_2$ or less would decrease from −0.12 W/m² under HIST to a maximum of −0.04 W/m² under SSP585. This calculation uses many simplifying assumptions, in particular that smaller magnitude eruptions all inject at the tropopause level under HIST conditions resulting in largely overestimating the forcing change as $SO_2$ injection heights can be higher and the stratospheric $SO_2$ inputs unaffected by the tropopause height increase. Overall, it is thus difficult to estimate whether the long-term net effect of climate–volcano feedback across all eruption types would be a forcing increase or decrease, and implementing a statistically realistic distribution of eruptions in a range of climate projections[68] is required to accurately quantify this.

Experiments similar to our study should also be conducted for extra-tropical eruptions, including effusive eruptions such as that of Laki in CE 1783–1784[69], to gain further understanding of how climate–volcano feedbacks depends on the eruption type. In addition to a future climate, climate–volcano feedbacks may also affect volcanic sulfate aerosol forcing under past climate conditions. Climate model simulations suggest there would be a relatively lower tropopause and slower Brewer–Dobson circulation in colder climates[70], and vice versa in warmer climates[71]. This in turn suggests that in past warm climates, such as the Eocene, the forcing associated with large-magnitude tropical eruptions was enhanced and that the forcing associated with moderate-magnitude tropical eruption was damped compared to today, and vice versa for past cold climates, such as the last glacial maximum. Lastly, our study demonstrates that accounting for changes in eruptive column dynamics is required to rigorously assess the impacts of climate change on volcanic radiative forcing.

Collaborative efforts between volcanologists and climate scientists are thus key to foster further progress in our understanding of interactions between the climate system and volcanoes. Some of our results are also of interest to the geo-engineering community, e.g., our finding that for a fixed $SO_2$ mass, a small reduction in the aerosol lifetime in UKESM1 causes a large enough reduction of the aerosol effective radius to increase radiative forcing. This provides a new perspective on how the efficiency of stratospheric aerosol geoengineering may be modulated by changes in atmospheric circulation driven by anthropogenic greenhouse gases emissions[47], tropospheric aerosols[72], or stratospheric aerosol geoengineering itself[33].

## Methods

**Interactive stratospheric aerosol model setup.** We use version 1.0 of the UK Earth System Model (UKESM1)[41] in an atmosphere-only configuration with version 11.2 of UM-UKCA[40,42], which is the atmospheric chemistry–aerosol–climate component of UKESM. As part of UM-UKCA, the GLOMAP-mode aerosol scheme simulates aerosol microphysical processes and calculates aerosol optical properties[40,48]. The resolution is 1.875° longitude by 1.25° latitude with 85 vertical levels extending from the surface to 85 km. This model is an interactive stratospheric aerosol model and, provided an initial emission of $SO_2$, it interactively simulates the chemical conversion into sulfate aerosol, the transport and loss of these aerosols, their interaction with radiation from the Sun and the Earth and the associated climate response[40]. UM-UKCA is the only model that has both an interactive OH cycle and an internally generated QBO among models with interactive stratospheric aerosols that contributed to the coordinated multimodel Tambora experiment of the Model Intercomparison Project on the climate response to Volcanic forcing (VolMIP)[46]. The tropospheric and stratospheric chemical and microphysical schemes have been shown to perform well compared to observations[42] and simulate sulfate aerosol properties and radiative forcing in general agreement with observations for the eruptions of Mt Agung (1963), El Chichón (1982), and Mt Pinatubo (1991), albeit with a downward adjustment of the injected $SO_2$ mass compared to observations[73]. For this study, compared to the CMIP6 configuration, SAOD and radiative forcing are determined by the stratospheric aerosol properties simulated by GLOMAP instead of the prescribed fields used in CMIP6 experiments[24], and the chemical change in $H_2SO_4$ is integrated as part of the aerosol routines. We conduct atmosphere-only time-slice experiments in this study, i.e., sea surface temperature, sea ice fraction and depth are prescribed, and forcing agents (e.g., carbon dioxide, methane) are also prescribed as a climatology. For the historical 1995 experiments (labeled HIST), these climatologies are derived from the 1990–2000 period as simulated in a UKESM1.0 historical run produced for CMIP6. For the SSP5 8.5 2095 experiments (labeled SSP5585), these climatologies are derived from the 2090–2100 period as simulated in a UKESM1.0 SSP5 8.5 run produced for CMIP6 ScenarioMIP[24]. For both scenarios, 15-years of spin-up were run in our time-slice atmosphere-only simulations starting from a set of initial conditions from a CMIP6 transient coupled run, followed by a 20-year control run without any volcanic eruptions.

**Eruptive column model setup.** We use the 1D eruptive column model of Degruyter and Bonadonna (2012)[43], whose main inputs are the mass eruption rate (also called eruption intensity, and referring to the total rate at which solid and gaseous products are expelled though the vent) and the atmospheric conditions at the vent location (Supplementary Fig. 1). This model solves equations governing the conservation of mass, momentum, and heat in the plume. The turbulent entrainment of atmospheric air into the plume is parameterized using two coefficients, the radial entrainment coefficient ($\alpha$), and the wind entrainment coefficient ($\beta$). The condensation of water vapor in the plume is governed by the condensation rate ($\lambda$). The main output of the model that we use is the neutral buoyancy level or height, i.e., the height at which the plume is as dense as the surrounding atmosphere and above which it will spread. The model also outputs the top plume height at which the plume momentum is exhausted. We assume that the neutral buoyancy level is the output most representative of the $SO_2$ injection height and refer to this height as the plume height or $SO_2$ injection height. The model does not treat the gas phase and solid phase of the plume independently and treats the plume as a monophasic fluid whose bulk properties are dependent on properties of the gas and solid phases, and the mass fraction of solids in the plume. The model employed has a level of complexity ideally suited for our study. In particular, in contrast to 0D eruptive column models (i.e., simple plume height scalings), they account for the role of vertically-varying atmospheric profiles in determining the height of the plume, which is key to understand the impact of climate change on $SO_2$ injection height. Despite being relatively simple relative to more complex 3D eruptive column models which can account e.g., for gas-particle coupling and rely on less simplistic closure for turbulence, 1D models are computationally inexpensive allowing us to extensively sample meteorological conditions simulated by UKESM1 and to conduct sensitivity test, which would be prohibitive in terms of computational cost with a 3D eruptive column model.

We use model parameter values that result in reasonable agreement with analog laboratory experiments and observations of historical eruptions[39] as well as a 3D eruptive column model[35]:

- $\alpha = 0.1$ for the radial entrainment coefficient and $\beta = 0.25$ for the wind entrainment coefficient, which result in relatively small rates of turbulent entrainment of the atmosphere into the plume and a relatively small impact of wind speed on the plume height.
- $\lambda = 10^{-6}\,s^{-1}$ for the condensation rate, which results in a negligible impact of water vapor condensation on the plume buoyancy flux and the height reached by the plume.

These parameters are subject to considerable uncertainties, in particular for $\beta$ and $\lambda$. We provide key sensitivity tests of the dependence of our plume height predictions in the HIST and SSP585 climate to these model parameters in Supplementary Fig. 3a, b (also see Discussion section), and extensive sensitivity tests are available in ref. [34].

Last, beyond the mass eruption rate and atmospheric profiles, the 1D eruptive column model also requires the following inputs:

- The vent altitude set to 1500 m, similar to that of Mount Pinatubo
- The exit gas mass fraction set to 5 wt% (weight total %)
- The exit temperature set to 1100 °C
- With the above parameters prescribed, varying the vent radius and exit velocity enable us to obtain the desired mass eruption rate; we fix the ratio of the vent radius to the square of exit velocity to 0.002. This ratio is directly proportional to the source Richardson number, i.e., the ratio of the plume source buoyancy flux to the plume source momentum flux, one of the key parameter governing the eruptive column stability (i.e., whether it collapses or rises as a buoyant plume).

These "secondary" inputs parameters may vary depending on the type of eruption. We provide key sensitivity tests of the dependence of our plume height predictions in the HIST and SSP585 climate to secondary model inputs parameters in Supplementary Fig. 3c, d, and extensive sensitivity tests are also available in ref. [34]. In particular, regardless of the values used for the secondary inputs parameters, our core results for plume height change hold: in SSP585 relative to HIST, higher mass eruption rates are required to reach the tropopause, and eruptive plumes reaching the stratosphere will see their height increase by ca. 1.5 km.

**Simulation of volcanic eruptions with our combined plume-aerosol-climate modeling framework.** When modeling the climate response to a volcanic eruption, interactive stratospheric aerosol studies typically prescribe an injected $SO_2$ latitude, longitude, mass, and altitude as the starting point of the simulation. Using the above approach does not allow the climate–volcano feedback to be fully understood as the $SO_2$ injection height may be affected by climate change[34,35] because atmospheric conditions (stratification, wind) exert an important control on the dynamics of eruptive columns. Here, we instead consider the starting point of the simulation to be the eruptive vent at the surface and prescribe the eruption intensity (also called mass eruption rate) instead of the $SO_2$ injection height. The height of $SO_2$ injection is calculated using the eruptive column model described above with core inputs of the mass eruption rate and the atmospheric conditions at the vent location simulated by UKESM1 at the beginning of the eruption day, which is also the first day of our 3-year simulations (Fig. 1). $SO_2$ is then distributed following a Gaussian distribution centered on this height with width 10% of the height, consistently with 3D eruptive column model simulations[35].

In an ideal setup, the eruptive column model would be run at every time step of the UKESM1 model during the eruption. However, such set-up would require full integration of the eruptive column model into the climate model which is beyond the scope of our study. Using a simplified approach, our study provides the first line of evidence that such integration could be valuable to accurately predict volcanic forcing in different climates. Our simplified approach mostly has two limitations. First, it neglects the variability of atmospheric conditions at sub-daily timescales during the eruption. Hourly atmospheric profiles were not outputted in our UKESM simulations, but using hourly profiles from the ERA5 reanalysis[74] at the location of Mount Pinatubo and running the eruptive column model employed for all July 1st profiles in the last 20 years, we find that over 85% of the variability in simulated plume height is associated with interannual variability (as opposed to hourly variability). We conclude that our design enables us to sample well the atmospheric conditions of each climate state. Second, our approach neglects the potential impacts of the early plume development on atmospheric conditions, which could in turn modulate the height reached by the eruptive column. The few studies that have quantified the local, instantaneous response of atmospheric conditions to volcanic eruptions suggest a warming temperature response near the plume top region for plumes composed mostly of $SO_2$, but a cooling temperature response for those composed mostly of ash[75]. Accounting for such effects would thus require ash to be co-emitted with $SO_2$ which is not currently possible with UKESM. However, atmospheric conditions will likely be affected below the spreading plume, downwind of the vent, and it thus remains unclear whether such effects would be of critical importance for modeling the column rise accurately.

**Experimental design**. Figure 1 and Table 1 summarize our experimental design. We ran experiments for two tropical eruption cases:

1. A moderate-magnitude eruption scenario with an intensity of $1.3 \times 10^7$ kg/s and a total mass of $SO_2$ of 1 Tg. This scenario is aimed to be representative of VEI 3–5 eruptions injecting on the order of 1 Tg of $SO_2$ or less in the upper-troposphere-lowermost stratosphere (under historical climate conditions) with a return frequency on the order of a year (e.g., Merapi 2010, Nabro 2011, Kelud 2014, Taal 2020).

2. A large-magnitude eruption scenario with an intensity of $2.7 \times 10^8$ kg/s and a total mass of $SO_2$ of 10 Tg. This scenario is aimed to be representative of VEI 5–6 eruptions injecting on the order of 10 Tg of $SO_2$ in the low-mid stratosphere (under historical climate conditions) with a return frequency on the order of decades (e.g., Agung 1963, El Chichon 1982, Mt. Pinatubo 1991).

Eruption intensities were chosen to obtain injection heights of ca. 16 and 21 km a.s.l. under HIST atmospheric conditions (Supplementary Fig. 2), and they differ by slightly more than a factor of 10 between the two eruption cases. In all experiments, the $SO_2$ injection occurs on July 1st in the model column containing the Mount Pinatubo location (15.1°N,120.4°E) with an $SO_2$ injection lasting 24 h. For each pair of eruption case and climate scenario (six pairs), we ran ten 3-year ensembles experiments with initial conditions extensively sampling $SO_2$ injection height and initial QBO phases (Fig. 2). In addition to the two climate scenarios mentioned previously, we run a third scenario (labeled SSP585_HIH, Historical Injection Height) where UKESM1 (atmosphere-only configuration) is run under the 2095 SSP5 8.5 forcings, but the $SO_2$ is injected at the heights obtained with atmospheric conditions from the historical 1995 run. The comparison between HIST, SSP585, and SSP585_HIH ensembles enables us to assess the role of changes in $SO_2$ injection height in modulating the sulfate aerosol cycle and radiative forcing response to the eruptions.

**Use of the FaIR model for estimating the surface temperature response**. The simulations conducted with UKESM1 are atmosphere-only simulations with prescribed sea surface temperatures so that we cannot directly diagnose the surface temperature response. To do so, we use the simple emission-based climate model Finite Amplitude Impulse Response (FaIR vn1.4)[51,52]. FaIR uses emissions of greenhouse gases and short-lived climate forcers to compute concentrations and radiative forcing time series, which are then converted to a global temperature anomaly. For each of our UKESM simulations, we ran a simulation using FaIR with the 3-year UKESM1 global post-eruption annual-mean TOA net forcing anomaly time series (Fig. 3d) and all other forcings as in 1850 preindustrial conditions as provided with the FaIR model (see also http://homepages.see.leeds.ac.uk/~mencsm/fair.htm). FaIR was additionally run without the extra volcanic forcing to compute the temperature anomaly due to the eruption. The obtained temperature anomalies in response to volcanic forcing are nearly independent of the underlying background conditions used in FaIR (we tested 1850 preindustrial, 2020 RCP4.5 and 2100 RCP8.5 climate conditions, with differences of less than 0.2% in simulated post-eruption temperature anomalies) which is why we provide all FaIR results using the same 1850 preindustrial background conditions.

**Anomalies, significance tests, and e-folding times**. All anomalies are calculated with respect to climatologies of the 20-year control runs with corresponding climate forcing, i.e., anomalies in HIST runs are calculated with respect to the 1995 historical control run, and anomalies in the SSP585 and SSP585_HIH are calculated with respect to the 2095 SSP5 8.5 control run. All significant tests performed are nonparametric Mann–Whitney $U$-tests (one-sided), testing the null hypothesis that a randomly selected value from a first population is larger than a randomly selected value from a second population. All significance tests are provided at the 95 and 80% level and, unless specified in the text, we discuss results at the 95% level. We provide three different e-folding times in this study (Figs. 4 and 6): the $SO_2$, total S and total tropical S burden e-folding times. Total S burden e-folding times were calculated using the total mass of sulfate in $SO_2$ and $H_2SO_4$ species, with the total tropical S burden restricted to 23.4°S–23.4°N and other e-folding times being for global burdens. The e-folding times were then simply obtained by fitting an exponential model of the form $a \times e^{-t/\tau}$ with $t$ the time in a month after the eruption, $a$ and $\tau$ the fit parameter with the latter being the e-folding time. Using the total S burden instead of the $H_2SO_4$ burden to calculate an aerosol e-folding time and quantify aerosol lifetime is more practical as the total S burden only decreases from the eruption onward.

**Back-of-the-envelope calculation for the net effect of climate–volcano feedbacks**. To provide an initial assessment of whether the increased forcing of large-magnitude eruptions or the decreased forcing of moderate-magnitude eruptions would dominate the net, 100-year time-averaged change in volcanic radiative forcing, we propose the following back-of-the-envelope calculation:

1. First, we partition the total amount of $SO_2$ emitted into the stratosphere into eruptions injecting less than and more than 3 Tg of $SO_2$. This 3 Tg threshold is midway (on a logarithmic scale) between the 1 Tg mass of our moderate-magnitude eruption case and the 10 Tg mass of our large-magnitude eruption case. For eruptions injecting >3 Tg $SO_2$ that have a small return period, we estimate their inputs from ice-core records of sulfate deposition[2]. We find that a total of 2243 Tg of $SO_2$ was injected over 2500 years by volcanic eruptions injecting >3 Tg $SO_2$. We hypothesize that this represents only stratospheric injections even though sulfate emitted into the troposphere may be deposited in polar ice-core for a volcano within close proximity of the poles (e.g., Iceland). On average, volcanic eruptions injecting over 3 Tg of $SO_2$ are thus associated with a flux of 90 Tg $SO_2$/century into the stratosphere. For eruptions injecting <3 Tg, which are poorly recorded in the ice-core archives, we use 1978–2015 satellite observations[7] and find that the total stratospheric $SO_2$ inputs from eruptions emitting <3 Tg $SO_2$ into the stratosphere is 14.66 Tg $SO_2$, corresponding to a flux of ca. 39 Tg $SO_2$ per century.

2. Second, we represent a given flux of volcanic $SO_2$ into the atmosphere per century, in Tg $SO_2$/century, by the number of eruptions similar to either our moderate-magnitude or large-magnitude case required to match the $SO_2$ flux per century. For eruptions injecting >3 Tg $SO_2$, we thus represent their stratospheric $SO_2$ inputs of 90 Tg $SO_2$/century as nine large-magnitude eruption occurring in a century. For eruptions injecting <3 Tg $SO_2$, we represent their stratospheric $SO_2$ inputs of 39 Tg $SO_2$/century as 39 moderate-magnitude eruption occurring in a century.

3. Third, we assume that the 3-year mean forcing obtained in Fig. 2 for single moderate-magnitude and large-magnitude eruption cases can be linearly superposed if multiple eruptions occur in a century, and that we can use our data on the first 3 post-eruption years to assess the total energy loss caused by an eruption. Consequently, for a single large-magnitude eruption and under HIST conditions, we assume that the 3-year mean forcing of 1.09 W/m$^2$ found would lead to a 100-year mean forcing of $1.09 \times 3/100 = 0.0327$ W/m$^2$. For a 100-year period under HIST conditions, we thus find that eruptions injecting >3 Tg $SO_2$ represented by nine large-magnitude eruption would exert a mean forcing of $9 \times 0.0327 = 0.29$ W/m$^2$. The same calculation under SSP585 conditions, for which we found a 3-year mean forcing of $-1.20$ W/m$^2$ for an individual large-magnitude eruption, leads to a 100-year mean forcing of 0.32 W/m$^2$. For eruptions injecting >3 Tg of $SO_2$, climate change would thus induce an additional $-0.03$ W/m$^2$ of volcanic forcing. Repeating the same calculation for eruptions injecting less than 3 Tg $SO_2$, we find a forcing of $-0.12$ W/m$^2$ under HIST conditions and $-0.04$ W/m$^2$ under SSP585, with climate change thus leading to a reduction of volcanic forcing by 0.08 W/m$^2$.

Despite the simplicity of our approach, the value we derive for large-magnitude type eruptions under HIST conditions is comparable to the latest estimate of long-term time-averaged volcanic forcing ($-0.29$ vs $-0.20$ W/m$^2$, see ref. [76]), and the value we derive for moderate-magnitude type eruptions under HIST conditions ($-0.12$ W/m$^2$) equals that found for the 2005–2015 period characterized by eruptions injecting less than 2 Tg of $SO_2$[17].

At first sight, our back-of-the-envelope calculation suggests that at centennial timescales, the forcing increase of large-magnitude type (injecting >3 Tg $SO_2$) eruptions will be ca. 3 times smaller than the forcing decrease associated with moderate-magnitude type (injecting <3 Tg $SO_2$) eruptions (0.03 vs 0.08 W/m$^2$). However, for our moderate-eruption case, the decrease in stratospheric aerosol burden by ca. 70% in our simulations represents an upper estimate as moderate-magnitude eruptions can inject $SO_2$ several kilometers above the tropopause in which case the future increase of the tropopause height would have a minimal effect on the stratospheric aerosol burden. Aubry et al. (2016)[34] estimated that the flux of $SO_2$ to the stratosphere from eruptions injecting 3s Tg $SO_2$ would decrease by 5% between the late 20th and late 21st century (for an upper-end greenhouse gas emission trajectory) which would make the forcing decrease for moderate-magnitude eruptions on the order of 0.005 W/m$^2$, much smaller than the forcing increase of 0.03 W/m$^2$ for large-magnitude eruptions. Clarifying the net effect of climate–volcano feedbacks will thus require further work including statistically realistic eruption distributions in terms of frequency, $SO_2$ mass, intensity, and location, as well as ocean–atmosphere coupled climate modeling to account for feedback related to ocean stratification[28].

## Data availability

The data that support the key findings in this study have been deposited in the Symplectic Elements data repository of the Cambridge University at https://doi.org/10.17863/CAM.66636. All other data are available from the corresponding author upon request.

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

## Acknowledgements

T.J.A. acknowledges support from the Royal Society through a Newton International Fellowship (grant number NIF\R1\180809), from the European Union's Horizon 2020 research and innovation program under the Marie Skłodowska-Curie grant agreement No 835939, and from the Sidney Sussex college through a Junior Research Fellowship. J.S.-S. is supported by NERC through the University of Cambridge ESS-DTP. A.S. acknowledges funding via the NERC V-PLUS project (NE/S00436X/1). L.R.M. and A.S. are funded by the UK Natural Environment Research Council (NERC) via the "Vol-Clim" grant (NE/S000887/1). J.H. and A.S. contribution benefitted from support by the NERC ADVANCE (Aerosol–cloud–climate interactions deduced using Degassing Volcanic Eruptions), grant NE/T006897/1. This work used the ARCHER UK National Supercomputing Service.

## Author contributions

T.J.A. designed the study, ran the UKESM1 experiments, analyzed the results, and wrote the manuscript with guidance from A.S. J.H. provided support for designing the study and analyzing the results. J.S.-S., L.R.M., and N.L.A. provided support for running the experiments, and L.R.M. also ran the FaIR model and contributed to the analysis of the results. All authors contributed to revising the manuscript.

## Competing interests

The authors declare no competing interests.
