## [Peer Review File · Nature Communications]

REVIEWER COMMENTS

Reviewer #1 (Remarks to the Author):

Summary:

Aubry et al. use the new UKESM to evaluate the influence of a warmer future climate on the radiative forcing from medium and large volcanic eruptions. They show that medium size eruptions will produce a much smaller radiative perturbation because the plume struggles to reach the stratosphere. In contrast for large eruptions several aspects of the warmer climate lead to a larger radiative forcing.

Recommendation:

The manuscript is well written and I appreciated the thorough and clear explanations of mechanisms presented. The results are important because they provide a robust benchmark for how volcanic eruptions will be altered by warming in the future. I recommend publication with minor corrections as detailed below.

Main comments:

I had one suggestion that may or may not be useful is to provide a figure or radiative forcing value that could be more easily understood by climatologists. Could you use your results to re-calculate the time-average radiative forcing in a RCP85 climate if the same eruptions occurred then as have occurred in the past 50-100 years?

Lines 322-323: these % changes are different from the abstract. Have I missed something here?

Lines 343-352: Could you also speculate on the role of climate-eruption interactions in general i.e. warm (Eocene) or cold (the last glacial)?

Lines 351: What about the role of tropospheric aerosols? This could also modulate the response (but not by as much as 40%)?

Line 356: "Earth climate sensitivity"
climate sensitivity has a very specific meaning related to CO₂, and I'm not sure you meant that here. How about 'the Earth System response' or something similar?

Line 356: "climate-volcano feedbacks could strongly modulate future climate projections"
I think this may be overstating it. How about "climate-volcano feedbacks will alter the decadal scale climate variability that is due to eruptions." or similar.

Minor comments:

Line 524: remove 'Discussions'.

Methods: I think you need a short paragraph describing the setup of the FAIR simulations.

Figure 1: I know it's a bit subjective but I think the legend would be much neater if it was placed outside of the figure.

Reviewer #2 (Remarks to the Author):

This study investigates climate change impacts on the volcanic sulfate aerosols life cycle and radiative forcing by employing three different models. The novel aspects originates from calculating the volcanic eruption height for a moderate (1 Tg SO₂) and large (10 Tg SO₂) eruption with a 1D eruption column model taking future atmospheric conditions into account, which then is applied to a complex aerosol-chemistry atmospheric model. Surface air temperatures changes are estimated with a simplified emission based climate model. This study contains novel aspects, which are of interest for a wider readership. However, the authors need to consider the following general and specific comments before publication in Nature Communications can be recommended.

General comments:

The paper has substantial caveats especially in the method and discussion parts as further described below, which needs to be addressed.

Methods:

The model approach and experimental setup needs further clarification. If I understand the method and manuscript description correctly, applied the authors a three-step approach. 1) UKESM1-AMIP was run with an eruption intensity at the vent at the surface releasing total mass flux (in kg/s of S or SO₂?) for HIST 1995 and SSP585 2090-2100 time slice conditions. I assume that the model was only run for a few hours or days? 2) The modelled atmospheric conditions were then used as input for the 1D volcanic plume model to calculate the corresponding eruption heights. 3) The derived eruption heights were then applied again to the UKESM1-AMIP experimental set ups injecting 1 Tg SO₂ and 10 Tg SO₂ between 14 and 23 km altitudes. Each simulation run for 3 years. Several questions arise:

- How long were the runs for 1) and how do the results look like for these first hours to days? I suggest adding a figure and details to the paper, as this is a key result.

-In Fig. 1 we can see 10 different eruption heights for each model experiment, so I assume that steps 1) and 2) were run 10x in line with step 3) as well?

Overall, Table 1 and the description of the model approach and experiments need to be updated including all experiments details. Right now, it is impossible to follow the study details.

-Details and explanations of the 1D eruption column model are missing although a key for the results of this paper. One has to look up earlier Aubry et al 2016 and 2019 papers to find these details.

-The climate volcanic response between HIST and SSP585 is a key result for this paper but what are actually the differences between the two scenarios? > Which scenarios are used for CO₂, CH₄ and CFCs for the future and past climate in detail? Why is there a higher OH in the future SSP585 scenario compared to the historical run (Figure S1); is that related to the forcings? What are the different QBO initial states? These details should be added to, e.g., Table 1.

-The simple emission based climate model FaIR for the SAT (Fig 8c) calculations need to be described. What was used as input for FaIR etc?

Discussion:

-The authors mentions the increase of the BDC/residual circulation in the stratosphere in a future climate as a "key driver of these changes" (line 325 and elsewhere) but the authors do not discuss the effects of volcanoes on the BDC/ residual circulation itself. Earlier studies showed an increase in the BDC/ residual circulation due to volcanic eruptions in the past, present and future climate (Toohey et al 2014 ACP, Muthers et al 2016 GRL and Garfinkel et al 2017 ACP). The authors could, f.e., check the differences of HIST-CTR and SSP585-CTR in contrast to HIST and the SSP585-HIST

difference (Figure S1g-h), which should give insights on the volcanic versus the climate BDC impacts. This needs to be addressed in the results and discussion.

-The role of water vapor phase changes is clearly mentioned as a challenge for the volcanic plume modelling but not for the aerosol chemistry atmosphere model. How is water vapour in the stratosphere treated? How is it changing in the future? Figure S1d actually shows H₂O for SSP585-HIST, which is not further described in the text. LeGrande et al (2016) discuss the role of atmospheric chemistry and in particular of stratospheric water vapor for modeling volcanic eruptions.

-The effects of an interactive ocean are missing; a caveat which is clearly stated by the authors. Thus, they have estimated surface temperature changes with the FaIR model. However, the use of the FaIR model is not mentioned in the abstract nor in the method part only in Figure 8. Next surface temperature changes are given in the abstract without mentioning the different model estimates. This needs to be taken care in the abstract, conclusions and method description.

- The discussion of future changes of the tropopause (Santer et al 2003) and the BDC (Butchart et al 2014) are based on "older" models and references. Is the new UKESM1-AMIP model but are also the new generation of CMIP6, ESM, CCMi models underlining these former cited results? I suggest adding new references here or a discussion of this point.

-A discussion of the performance of the aerosol module of the UKESM1 model, see e.g. the VolMIP special issue, is missing. How is the UKESM1 performance compared to other aerosol chemistry climate models, can we trust it? What are the uncertainties and limitations?

Title and abstract:

As the title specifies, new aspects of the paper include the aerosol life cycle and the radiative forcing. However, the abstract highlights only the temperature changes in the stratosphere, troposphere and surface specifically. Here the % changes of the aerosol size and radiative forcing should be added as well before coming to the temperature changes.

Where do the 52%, 51% and 15% changes originate? Why do you call them "strongly"? Inconsistent use of "strong" model response throughout the results. I suggest deleting the word "strongly" in the title.

Introduction:

-The introduction lists three different climate impacts on volcanic eruptions but neglects the glaciation-interglaciations climate impacts on volcanic eruption frequency (Kutterolf et al, 2013).

-Lines 130-131: "...the decreasing stratospheric stratification." Why and what is causing it? This is a key aspect of this current paper and needs to be explained here without the necessity for checking earlier companions papers.

Supplementary Information:

Table S1: The John Staunton-Sykes et al (in prep.) paper is not available to the reviewers (and readers) and thus should either be provided or deleted in Table S1 and corresponding text.

Specific comments

-Lines 35-37: "...also effects key modes of climate variability such as the ..." ENSO, NAO, the monsoon and AMOC. This is still under discussion in the scientific community and it is not clear at all. This sentence needs a reformulation.

-Line 42: Reference 17 (Hansen et al 1978) is an early and original reference but does not fit exactly to the statement here "has been a major research topic for decades". An additional ref is needed to underline this point.

- Line 56: Reference 23 is not a peer-reviewed article and needs to be highlighted as such.
- Line 97: double "that"
- Lines 108-109: Where do these 1.3×10^7 kg/s and 2.7×10^8 kg/s numbers come from?
- Line 124: "...with much smaller injection height of 14-15 km..." smaller than what?
- Lines 169-183: "TOA forcing" and "surface forcing" should be "TOA radiative forcing" and "surface radiative forcing". How much is CO₂, CH₄ etc changing between HIST and SSP585?
- Line 190: Why do you have higher OH value in the future? Needs further explanation.
- Line 213: Change to: "The sulfuric acid aerosols spread faster than..."
- Line 227: Unclear, for which experiment?
- Lines 307-308: What about the role of the QBO here?
- Line 310: "volcanic ash or halogens" are cited wrt Muser et al 2020 ACP (on the Raikoke eruption), which only investigates volcanic ash.
- Line 332: Add Marshall et al 2018 and Brenna et al 2020 (both VolMIP ACP special issue) for the aerosol model discrepancies here as well.
- Nomenclature "large" and "strong" eruptions: Suggest using just one of the terms in the paper (see Figure 1 versus Figure 2).

Figure 3: Why are you not showing the large eruption figure in the Supplementary Information as well?

References:

- Brenna, H., Kutterolf, S., Mills, M. J., and Krüger, K.: The potential impacts of a sulfur- and halogen-rich supereruption such as Los Chocoyos on the atmosphere and climate, *Atmos. Chem. Phys.*, 20, 6521–6539, <https://doi.org/10.5194/acp-20-6521-2020>, 2020.
- Garfinkel, C. I., Aquila, V., Waugh, D. W., and Oman, L. D.: Time-varying changes in the simulated structure of the Brewer–Dobson Circulation, *Atmos. Chem. Phys.*, 17, 1313–1327, <https://doi.org/10.5194/acp-17-1313-2017>, 2017.
- Kutterolf et al 2013 A detection of Milankovitch frequencies in global volcanic activity, *Geology* (2013) 41 (2): 227–230 <https://doi.org/10.1130/G33419.1>
- LeGrande A.N., K. Tsigaridis, and S.E. Bauer, 2016: Role of atmospheric chemistry in the climate impacts of stratospheric volcanic injections. *Nature Geosci.*, 9, no. 9, 652-655, [doi:10.1038/ngeo2771](https://doi.org/10.1038/ngeo2771).
- Marshall, L., A. Schmidt, M. Toohey, K.S. Carslaw, G.W. Mann, M. Sigl, M. Khodri, C. Timmreck, D. Zanchettin, W. Ball, S. Bekki, J.S.A. Brooke, S. Dhomse, C. Johnson, J.-F. Lamarque, A. LeGrande, M.J. Mills, U. Niemeier, J.O. Pope, V. Poulain, A. Robock, E. Rozanov, A. Stenke, T. Sukhodolov, S. Tilmes, K. Tsigaridis, and F. Tummon, 2018: Multi-model comparison of the volcanic sulfate deposition from the 1815 eruption of Mt. Tambora. *Atmos. Chem. Phys.*, 18, 2307–2328, [doi:10.5194/acp-18-2307-2018](https://doi.org/10.5194/acp-18-2307-2018).
- Muser, L. O., Hoshyaripour, G. A., Bruckert, J., Horváth, Á., Malinina, E., Wallis, S., Prata, F. J., Rozanov, A., von Savigny, C., Vogel, H., and Vogel, B.: Particle aging and aerosol–radiation interaction affect volcanic plume dispersion: evidence from the Raikoke 2019 eruption, *Atmos.*

Chem. Phys., 20, 15015–15036, <https://doi.org/10.5194/acp-20-15015-2020>, 2020.

Muthers, S., A. Kuchar, A. Stenke, J. Schmitt, J. G. Anet, C. C. Raible, and T. F. Stocker (2016), Stratospheric age of air variations between 1600 and 2100, *Geophys. Res. Lett.*, 43, doi:10.1002/2016GL068734.

Toohey, M., Krüger, K., Bittner, M., Timmreck, C., and Schmidt, H.: The impact of volcanic aerosol on the Northern Hemisphere stratospheric polar vortex: mechanisms and sensitivity to forcing structure, *Atmos. Chem. Phys.*, 14, 13063–13079, <https://doi.org/10.5194/ac>

Reviewer comments are in italic and blue; our responses are in black. All changes to our original submission have been tracked using Microsoft Word track changes tool, except for reference numbers and section.

Reviewer 1

Summary:

Aubry et al. use the new UKESM to evaluate the influence of a warmer future climate on the radiative forcing from medium and large volcanic eruptions. They show that medium size eruptions will produce a much smaller radiative perturbation because the plume struggles to reach the stratosphere. In contrast for large eruptions several aspects of the warmer climate lead to a larger radiative forcing.

Recommendation:

The manuscript is well written and I appreciated the thorough and clear explanations of mechanisms presented. The results are important because they provide a robust benchmark for how volcanic eruptions will be altered by warming in the future. I recommend publication with minor corrections as detailed below.

We thank the reviewer for their enthusiastic evaluation of our paper and for the very useful comments on our manuscript.

Main comments:

I had one suggestion that may or may not be useful is to provide a figure or radiative forcing value that could be more easily understood by climatologists. Could you use your results to re-calculate the time-average radiative forcing in a RCP85 climate if the same eruptions occurred then as have occurred in the past 50-100 years?

We thank the reviewer for their important suggestion. We chose to not provide such a figure in our initial submission because we find that climate-volcano feedbacks depend on eruption source parameters, so that the right approach to this problem is to conduct simulations with a statistically realistic distribution of eruptions in different background climates. However, we made a back-of-the-envelope calculation detailed in a new subsection in Methods and summarized its results at the end of the discussion section:

“A back-of-the envelope calculation (see Methods) suggests that the long-term radiative forcing from eruptions injecting on the order of 10Tg SO₂ or more would increase (in magnitude) from -0.29 W/m² under HIST conditions to -0.32 W/ m² under SSP585 conditions, whereas that from eruptions injecting on the order of 1 Tg SO₂ or less would decrease from -0.12 W/ m² under HIST to -0.04W/ m² at most under SSP585. This calculation uses many simplifying assumptions, in particular that smaller magnitude eruptions all inject at the tropopause level under HIST conditions resulting in largely overestimating the forcing change as SO₂ injection heights can be higher and the stratospheric SO₂ inputs unaffected by the tropopause height increase. Overall, it is thus difficult to estimate whether the net – i.e. across all eruption types – effect of climate-volcano feedback would be a forcing increase or decrease, and implementing a statistically

realistic distribution of eruptions in a range of climate projections is required to accurately quantify this. “

Lines 322-323: these % changes are different from the abstract. Have I missed something here?

Thanks for pointing this out. The percentage changes provided in the abstract are referring to 3-year averages, whereas those previously given lines 322-323 were for the first post-eruption year. We now give percentage changes for the 3-year average in the main text as well to avoid confusion.

Lines 343-352: Could you also speculate on the role of climate-eruption interactions in general i.e. warm (Eocene) or cold (the last glacial)?

We thank the reviewer for their suggestion and have added the following sentences to the last paragraph:

“In addition to a future climate, climate-volcano feedbacks may also affect volcanic aerosol forcing under past climate conditions. Climate model simulations suggest there would be a lower tropopause and slower Brewer–Dobson circulation in colder climates (Wang et al., 2020), and vice versa in warmer climates (Szopa et al., 2019) than today. This in turn suggests that in past warm climates, such as the Eocene, the forcing associated with large-magnitude tropical eruptions was enhanced and that the forcing associated with moderate-magnitude tropical eruption was damped compared to today, and vice versa for past cold climates, such as the last glacial maximum.”

Lines 351: What about the role of tropospheric aerosols? This could also modulate the response (but not by as much as 40%)?

Non-volcanic tropospheric aerosols are explicitly included in our climate simulations through the SSP scenario framework. Even though we expect that they play a negligible role, the reviewer has a point and we now included the following sentences in the discussion:

“Another factor that could affect the climate response to volcanic eruptions is the tropospheric aerosol background state: using the previous generation (CMIP5) of model (HadGEM2-ES) and previous mid-high end emission scenario (Representative Concentration Pathway 6.0), Hopcroft et al. (2017) showed that an increase in tropospheric aerosol between a pre-industrial climate state and year 2045 results in an increased planetary albedo and reduced radiative forcing and surface cooling. Such a feedback mechanism would be accounted for in our simulations, but we find a minor decrease (as opposed to the increase in Hopcroft et al. (2017)) in tropospheric aerosol optical depth at 550nm between our two chosen climate states (-2% from HIST to SSP585, Table 1). As such, we expect the change in tropospheric aerosol to play a negligible role in our results when compared to the processes governing the stratospheric aerosol life cycle.”

Line 356: "Earth climate sensitivity"

climate sensitivity has a very specific meaning related to CO₂, and I'm not sure you meant that here. How about 'the Earth System response' or something similar?

We recognise that there are a few different definitions of climate sensitivity such as the equilibrium climate sensitivity that can be diagnosed from step increases in CO₂ and transient climate sensitivity that

is frequently diagnosed from transient climate change experiments - both are normally quoted in terms of K/Wm². If increased CO₂ leads to a change in volcanic forcing and associated surface cooling, the idea that these feedbacks affect Earth climate sensitivity appears justified. However, as our study does not allow us to quantify such an effect yet, we removed the mention of climate sensitivity to avoid any confusion.

Line 356: "climate-volcano feedbacks could strongly modulate future climate projections"

I think this may be overstating it. How about "climate-volcano feedbacks will alter the decadal scale climate variability that is due to eruptions." or similar.

We agree that this sentence was overstated and we have removed the word "strongly". The sentence now reads:

"However, the large enhancement obtained suggests that climate-volcano feedbacks could modulate the decadal scale climate variability driven by explosive eruptions and future climate projections."

Even though our work focuses on two idealized case study eruptions, volcanic eruptions large enough to be climatically relevant occur on a yearly basis. As such, it is reasonable to speculate on the impact of climate-volcano feedbacks on future climate projections.

Minor comments:

Line 524: remove 'Discussions'.

We have now removed it.

Methods: I think you need a short paragraph describing the setup of the FAIR simulations.

This was indeed an important omission and a dedicated section has been added to the Methods.

Figure 1: I know its a bit subjective but I think the legend would be much neater if it was placed outside of the figure.

We have followed this suggestion.

Reviewer 2

This study investigates climate change impacts on the volcanic sulfate aerosols life cycle and radiative forcing by employing three different models. The novel aspects originates from calculating the volcanic eruption height for a moderate (1 Tg SO₂) and large (10 Tg SO₂) eruption with a 1D eruption column model taking future atmospheric conditions into account, which then is applied to a complex aerosol-chemistry atmospheric model. Surface air temperatures changes are estimated with a simplified emission based climate model. This study contains novel aspects, which are of interest for a wider readership. However, the authors need to consider the following general and specific comments before publication in Nature Communications can be recommended.

General comments:

The paper has substantial caveats especially in the method and discussion parts as further described below, which needs to be addressed.

We thank the reviewer for their encouraging evaluation of our paper and for the very useful comments on our manuscript. Specifically, and as outlined below, we have now addressed the caveats that the reviewer quite rightfully highlighted by revising several aspects of the manuscript including the description of our methodology.

Methods:

The model approach and experimental setup needs further clarification. If I understand the method and manuscript description correctly, applied the authors a three-step approach. 1) UKESM1-AMIP was run with an eruption intensity at the vent at the surface releasing total mass flux (in kg/s of S or SO₂?) for HIST 1995 and SSP585 2090-2100 time slice conditions. I assume that the model was only run for a few hours or days? 2) The modelled atmospheric conditions were then used as input for the 1D volcanic plume model to calculate the corresponding eruption heights. 3) The derived eruption heights were then applied again to the UKESM1-AMIP experimental set ups injecting 1 Tg SO₂ and 10 Tg SO₂ between 14 and 23 km altitudes. Each simulation run for 3 years. Several questions arise:

The reviewer very correctly points out that the details of our experimental design were unclear in the main text. We have clarified these details by doing the following:

- A new figure 1 now gives the reader an overview of the experimental setup
- The Methods section now includes a subsection dedicated to the eruptive column model (also termed as plume model), with three new SI figures providing details on simulations conducted and sensitivity tests
- A new sub-section on the eruptive column model explains in detail as to how the eruptive column model and climate model were combined to produce our results
- Table 1 includes more details on the climate scenario used

The lack of clarity in the initial submission has, unfortunately, led to some confusion related to the first step. The plume and climate models are not coupled, i.e. we did not implement a 1D eruptive column model routine in the climate model. Instead, they are run separately but in series. The eruptive column model is run using atmospheric conditions extracted from the restart file (i.e. initial conditions) from each climate model simulations, themselves taken from the 20 year control simulation (without eruptions). The plume height obtained from the eruptive column model simulations are then used to define the SO₂ injection height for climate model simulations. These climate model simulations utilise the same restart file as was used for the eruptive column model simulations. In each of the climate model simulations, the eruption occurs on the very first day, meaning that the atmospheric conditions are exactly the same as in the eruptive column model at the beginning of the eruption day. A more realistic design would include a full coupling between the eruptive column model and the climate model, where atmospheric conditions to the eruptive column model are provided at each time step by the climate model, and are also influenced by the early development of the plume. However, such coupling is complex as it would require the development and implementation of a new routine in the climate model and we expect that it would make minimal difference to our results as the hourly variability in atmospheric conditions result in smaller plume height variations than the daily-inter-annual

variability which is accounted for in our approach. We hope that the new figure 1 and method text fully clarify our design.

- How long were the runs for 1) and how do the results look like for these first hours to days? I suggest adding a figure and details to the paper, as this is a key result.

We believe that this comment stems from the misunderstanding of our experimental design, now clarified above. There are no “runs for 1)” as described by the reviewer. However, we have included additional figures (S1-S2) to show the atmospheric profiles extracted from the restart files of each ensemble member simulation, and the corresponding relationship between injection height and mass eruption rate as simulated by the eruptive column model.

-In Fig. 1 we can see 10 different eruption heights for each model experiment, so I assume that steps 1) and 2) were run 10x in line with step 3) as well?

Yes, all experiments were repeated 10 times each with different initial conditions to assess uncertainties. This is hopefully made clear by Figure 1 and we note that the ensemble size was previously provided as well in Table 1 and in section “Changes in SO₂ injection height”

Overall, Table 1 and the description of the model approach and experiments need to be updated including all experiments details. Right now, it is impossible to follow the study details.

We provide a more detailed answer to that comment in our replies to specific comments below. We hope that the new Figure 1 clarifies our overall workflow and how the eruptive column model and climate model were combined.

-Details and explanations of the 1D eruption column model are missing although a key for the results of this paper. One has to look up earlier Aubry et al 2016 and 2019 papers to find these details.

We indeed relied too much on these previous studies and thank the reviewer for pointing this out. We now provide a detailed description of the 1D eruptive column model used in this work, in a new methods section in the manuscript. In particular, we now highlight the model parameters and inputs in addition to the eruption intensity and atmospheric conditions. This paragraph is accompanied by 3 new SI figures:

- Figure S1 provides the atmospheric profiles from UKESM1 used to run the eruptive column model
- Figure S2 shows the eruptive column model prediction for injection height as a function of the eruption intensity for atmospheric profiles shown in Figure S1. In particular, it clearly highlights that plumes previously only reaching the upper troposphere (under HIST) see their height decrease in SSP585, plumes previously reaching the vicinity of the tropopause we see their height remain unchanged and that plumes reaching the stratosphere see their height increase.
- Figure S3 shows how results from Figure S2 are sensitive to a range of model parameters and model inputs that were fixed for our study.

For a more extensive discussion into the uncertainties and sensitivities of the eruptive column modelling approach used in this work as well as for a comparison with a 3D eruptive column model result, we continue to direct the reader towards previous studies (Aubry et al. 2016, 2019).

-The climate volcanic response between HIST and SSP585 is a key result for this paper but what are actually the differences between the two scenarios? > Which scenarios are used for CO₂, CH₄ and CFCs for the future and past climate in detail? Why is there a higher OH in the future SSP585 scenario compared to the historical run (Figure S1); is that related to the forcings? What are the different QBO initial states? These details should be added to, e.g., Table 1.

Some differences between the HIST and SSP585 scenarios were indeed shown in the previous Figure S1 (now S4), but we have now expanded Table 1 to contain the following information:

- global mean sea surface, air surface and 50hPa temperature
- global mean CO₂, CH₄, O₃ Cl and Br concentrations
- global mean tropospheric and stratospheric AOD at 550nm

All QBO initial states are provided in Figure 2 and in the data repository accompanying this paper. The higher OH concentrations are driven by higher stratospheric water vapor concentration.

-The simple emission based climate model FaIR for the SAT (Fig 8c) calculations need to be described. What was used as input for FaIR etc?

Additional details on FaIR were lacking in our initial submission and we thank the reviewer for pointing that out. We have now added a Method subsection devoted to the FaIR model. In particular, we explain that we used pre-industrial 1850 conditions to run FaIR since temperature anomalies were nearly independent from the background (i.e. non-volcanic) forcings used in FaIR, with maximum differences in post-eruption temperature anomalies of 0.2% when using pre-industrial 1850 conditions vs 2100 RCP8.5 conditions.

Discussion:

-The authors mentions the increase of the BDC/residual circulation in the stratosphere in a future climate as a “key driver of these changes” (line 325 and elsewhere) but the authors do not discuss the effects of volcanoes on the BDC/residual circulation itself. Earlier studies showed an increase in the BDC/residual circulation due to volcanic eruptions in the past, present and future climate (Toohey et al 2014 ACP, Muthers et al 2016 GRL and Garfinkel et al 2017 ACP). The authors could, f.e., check the differences of HIST-CTR and SSP585-CTR in contrast to HIST and the SSP585-HIST difference (Figure S1g-h), which should give insights on the volcanic versus the climate BDC impacts. This needs to be addressed in the results and discussion.

We thank the reviewer for pointing out an interesting line of result discussion, which we had previously overlooked. We have now done this additional analysis and added a full paragraph devoted to the response of age of air and circulation to the large-magnitude eruption in our simulations, along with a new figure S8 showing the response of stratospheric age of air (note that the response of the winter polar vortex is still shown in Figure S6). The new paragraph is at the end of our result section:

“Lastly, some of the mechanisms via which climate change affects the volcanic sulfate aerosol cycle and forcing are also affected by the volcanic aerosol themselves. For example, the Brewer-Dobson circulation is accelerated by both anthropogenic greenhouse gas emissions⁴⁷ and volcanic aerosol from tropical eruptions^{55,56}. Our own experiments are consistent with an accelerated circulation following the large-magnitude eruption, with a decrease of stratospheric age of air in the Northern Hemisphere in all scenarios (Figure S8). Whereas the forcing and stratospheric warming is enhanced in the future climate, we find more pronounced age of air anomalies in the HIST scenario. The volcanically-forced acceleration of the Brewer-Dobson circulation, as well as the strengthening of the winter polar vortex, have been linked to enhanced wave propagation as a result of aerosol heating and the increase of the meridional stratospheric temperature gradient⁵⁷. Our results are consistent with this picture as the HIST scenario is characterized by a slower transport to high-latitude of the aerosol cloud (Figure 6.a-f) and a stronger meridional stratospheric temperature gradient (Figure 9.d-f). This in turn explains both a stronger acceleration of the Brewer-Dobson circulation (Figure S8) and strengthening of the polar vortex (Figure S6) compared to the SSP585 and SSP585_HIH scenarios. The eruption-induced change in the age of air (ca. -0.25 years between 20 and 30km altitude, Figure S8) is relatively small compared to the difference between the two climate scenarios considered (ca. -1 year, Figure S4.g-h).”

-The role of water vapor phase changes is clearly mentioned as a challenge for the volcanic plume modelling but not for the aerosol chemistry atmosphere model. How is water vapour in the stratosphere treated? How is it changing in the future? Figure S1d actually shows H2O for SSP585-HIST, which is not further described in the text. LeGrande et al (2016) discuss the role of atmospheric chemistry and in particular of stratospheric water vapor for modeling volcanic eruptions.

We thank the reviewer for pointing this omission and now discuss this limitation of our design where we also mention limitations related to the fact we do not co-inject ash and halogen with SO₂. We now also make use of previous Figure S1.e-f (now S4.e-f).

“Previous studies have shown that the co-injection of other volcanic products, including ash (Zhu et al. 2020), halogens (Staunton-Sykes et al., under review), and water (Legrande et al. 2016), would affect the aerosol cycle including the SO₂ lifetime or the self-lofting of the aerosol cloud. The co-injection of these products may also make for a more complex climate-volcano feedback picture, e.g. increased atmospheric humidity (Figure S4.e-f) may result in larger stratospheric injection of water by a volcanic plume via the turbulent entrainment of tropospheric water vapor into the rising plume.”

-The effects of an interactive ocean are missing; a caveat which is clearly stated by the authors. Thus, they have estimated surface temperature changes with the FaIR model. However, the use of the FaIR model is not mentioned in the abstract nor in the method part only in Figure 8. Next surface temperature changes are given in the abstract wo mentioning the different model estimates. This needs to be taken care in the abstract, conclusions and method description.

We have now added a subsection devoted to FaIR in methods and clearly mention it in conclusions. The abstract is restricted to 150 words which does not allow us to mention the use of FaIR without removing other information which we deem more important for the abstract.

- The discussion of future changes of the tropopause (Santer et al 2003) and the BDC (Butchart et al 2014) are based on “older” models and references. Is the new UKESM1-AMIP model but are also the new generation of CMIP6, ESM, CCM1 models underlining these former cited results? I suggest adding new references here or a discussion of this point.

We added a new reference with CMIP6 models for the tropopause height changes:

Griffiths, P. T., Murray, L. T., Zeng, G., Shin, Y. M., Abraham, N. L., Archibald, A. T., Deushi, M., Emmons, L. K., Galbally, I. E., Hassler, B., Horowitz, L. W., Keeble, J., Liu, J., Moeini, O., Naik, V., O'Connor, F. M., Oshima, N., Tarasick, D., Tilmes, S., Turnock, S. T., Wild, O., Young, P. J., and Zanis, P.: Tropospheric ozone in CMIP6 simulations, *Atmos. Chem. Phys.*, 21, 4187–4218, <https://doi.org/10.5194/acp-21-4187-2021>, 2021.

We could not find a CMIP6 reference showing how the BDC changes in the future across multiple models and welcome any suggestion, but this is a robust feature of climate projections and the cited review by Butchart is relatively recent.

Both a tropopause height increase and an acceleration of the BDC are simulated by UKESM1 as evident from our results.

-A discussion of the performance of the aerosol module of the UKESM1 model, see e.g. the VolMIP special issue, is missing. How is the UKESM1 performance compared to other aerosol chemistry climate models, can we trust it? What are the uncertainties and limitations?

The reviewer’s point is an important one. Note that the VolMIP core experiments do not aim to evaluate models with interactive stratospheric aerosols (volcanic forcing is prescribed in VolMIP). The “VolMIP-Tambora Interactive Stratospheric Aerosol ensemble” (Clyne et al., 2021) compared results from several models with interactive stratospheric aerosols but does not evaluate these models. However, Clyne et al. (2021) highlight that among models with interactive stratospheric aerosol capability, UM-UKCA is the only model that has both an internally generated QBO and interactive OH. The most relevant intercomparison exercise for aerosol module evaluation is the interactive stratospheric aerosol model intercomparison project (ISA-MIP) and we have summarized the recent related results of Dhomse et al. (2020) for the model we use:

“UM-UKCA is the only model that has both an interactive OH cycle and an internally generated QBO among models with interactive stratospheric aerosols that took part to the coordinated multimodel Tambora experiment of the Model Intercomparison Project on the climate response to Volcanic forcing (VolMIP)⁴⁶. The tropospheric and stratospheric chemical and microphysical schemes have been shown to perform well compared to observations⁴² and simulates sulfate aerosol properties and radiative forcing in general agreement with observations for the eruptions of Mt Agung (1963), El Chichón (1982), and Mt Pinatubo (1991), albeit with a downward adjustment of the injected SO₂ mass compared to observations⁷⁰. “

The downward adjustment of the injected SO₂ mass is required by other aerosol-chemistry-climate models (e.g. Mills et al. 2016) and is thought to be the consequence of a missing SO₂ removal processes in these models, with scavenging by co-emitted ash being a leading suspect (e.g. Zhu et al. 2020). At the moment we are not aware of any other published ISA-MIP results. Further efforts to validate the capability of the fully coupled version of UKESM1 to simulate volcanic aerosols, their radiative forcing

and the climate response are ongoing but the results are not published yet. Our original submission also refers to more general evaluations of UKESM1, in particular the aerosol component and stratospheric dynamics (Archibald et al., 2020).

Title and abstract:

As the title specifies, new aspects of the paper include the aerosol life cycle and the radiative forcing. However, the abstract highlights only the temperature changes in the stratosphere, troposphere and surface specifically. Here the % changes of the aerosol size and radiative forcing should be added as well before coming to the temperature changes.

Where do the 52%, 51% and 15% changes originate? Why do you call them “strongly”? Inconsistent use of “strong” model response throughout the results. I suggest deleting the word “strongly” in the title.

We have reduced the abstract text to include the % change in forcing as suggested. We could not add the % change in aerosol size due to length restriction for the abstract. The % changes for temperature are referring to 3-year averages, whereas those previously given in the text were for the first post-eruption year. We now give percentage changes for the 3-year average in the main text as well to avoid confusion. We have removed the word “strongly” in the title, abstract and also removed all unnecessary use of it in the main text.

Introduction:

-The introduction lists three different climate impacts on volcanic eruptions but neglects the glaciation-interglaciations climate impacts on volcanic eruption frequency (Kutterolf et al, 2013).

This type of feedback is the first category mentioned in our introduction (“1.The first category – the only one that has been subject to significant research efforts – relates to how changing climatic conditions affect the spatial, temporal and magnitude distribution of explosive eruptions. In particular, there is substantial evidence that eruption frequency and magnitude increase following the deglaciation of ice-covered volcanoes.”). We have used different references as the one suggested by the reviewer but we think they are appropriate and that no further reference is required given the limit of 70 references for Nature Communications.

-Lines 130-131: “...the decreasing stratospheric stratification.” Why and what is causing it? This is a key aspect of this current paper and needs to be explained here wo the necessity for checking earlier companions papers.

We have now included a figure showing both the temperature profiles and the stratification profile (Brunt-Vaisala frequency) at the location of Mount Pinatubo (Figure S1). The changes in stratification are a consequence of changes in the vertical temperature gradients caused by the dependence of temperature trends on altitude.

Supplementary Information:

Table S1: The John Staunton-Sykes et al (in prep.) paper is not available to the reviewers (and readers) and thus should either be provided or deleted in Table S1 and corresponding text.

We have deleted Table S1 and the corresponding text as suggested by the reviewer.

Specific comments

-Lines 35-37: "...also effects key modes of climate variability such as the ..." ENSO, NAO, the monsoon and AMOC. This is still under discussion in the scientific community and it is not clear at all. This sentence needs a reformulation.

We agree that our formulation should have been more careful, in particular for the ENSO response. We have edited this sentence to read:

"Large-magnitude eruptions may also affect key modes of climate variability such as the North Atlantic Oscillation¹¹, tropical monsoons^{12,13}, and the Atlantic meridional overturning circulation¹⁴, and could affect the El Niño Southern Oscillation although the sign and magnitude of the response is debated^{15,16}."

-Line 42: Reference 17 (Hansen et al 1978) is an early and original reference but does not fit exactly to the statement here "has been a major research topic for decades". An additional ref is needed to underline this point.

We have added the following reference to support this statement:

Rampino, M. R., & Self, S. (1982). Historic eruptions of Tambora (1815), Krakatau (1883), and Agung (1963), their stratospheric aerosols, and climatic impact. *Quaternary Research*, 18(2), 127-143, [https://doi.org/10.1016/0033-5894\(82\)90065-5](https://doi.org/10.1016/0033-5894(82)90065-5)

-Line 56: Reference 23 is not a peer-reviewed article and needs to be highlighted as such.

A peer reviewed article in line with this USGS communication has now been published and we replaced the reference accordingly:

Patrick, M.R., et al. The cascading origin of the 2018 Kīlauea eruption and implications for future forecasting. *Nat Commun* 11, 5646 (2020). <https://doi.org/10.1038/s41467-020-19190-1>

-Line 97: double "that"

We removed one of the occurrences.

-Lines 108-109: Where do these 1.3 x 10⁷ kg/s and 2.7 x 10⁸ kg/s numbers come from?

Figure S2 now gives more context on these numbers; they were simply chosen to obtain injection heights of ca. 16km a.s.l. and 21 km a.s.l. for the historical climate.

-Line 124: "...with much smaller injection height of 14-15 km..." smaller than what?

We meant smaller than all other injection heights for the SSP585 scenario, which is now clarified.

- Lines 169-183: "TOA forcing" and "surface forcing" should be "TOA radiative forcing" and "surface radiative forcing". How much is CO₂, CH₄ etc changing between HIST and SSP585?

We have corrected these throughout the manuscript. Changes in CO₂, CH₄ and CFCs are now included in Table 1.

-Line 190: Why do you have higher OH value in the future? Needs further explanation.

Increasing water vapor concentrations drive higher OH in the future. We have edited the corresponding sentence to read:

"This change is likely related to higher OH concentrations in the future (Figure S4.c,d, themselves caused by increasing stratospheric water vapor concentrations, Figure S4.e,f), and consequently faster conversion of SO₂ into sulfuric acid (H₂SO₄) via gas phase oxidation"

- Line 213: Change to: "The sulfuric acid aerosols spread faster than..."

Done.

-Line 227: Unclear, for which experiment?

We have specified the experiments (larger effective radius in HIST compared to SSP585, and SSP585 compared to SSP585_HIH).

-Lines 307-308: What about the role of the QBO here?

We focused on the three primary effects that would govern how much of the aerosol will end up in the stratosphere. However, the reviewer is correct that QBO may further modulate this rise and we have added the sentence:

"The QBO phase may also modulate the sulfur cloud lofting (Punge et al. 2009)."

after discussing self-lofting caused by absorption.

-Line 310: "volcanic ash or halogens" are cited wrt Muser et al 2020 ACP (on the Raikoke eruption), which only investigates volcanic ash.

Thanks for pointing this mistake out. We have now replaced this reference by:

Staunton-Sykes, J., Aubry, T. J., Shin, Y. M., Weber, J., Marshall, L. R., Abraham, N. L., Schmidt, A., and Archibald, A.: Co-emission of volcanic sulfur and halogens amplifies volcanic effective radiative forcing, Atmos. Chem. Phys. Discuss. [preprint], <https://doi.org/10.5194/acp-2020-1110>, in review, 2020.

-Line 332: Add Marshall et al 2018 and Brenna et al 2020 (both VolMIP ACP special issue) for the aerosol model discrepancies here as well.

Even though these are good suggestions and we would have liked to acknowledge more studies in many places in the manuscript, we already cite two references for this statement that are used elsewhere in the manuscript, and we have exceeded the limit of 70 references for Nature Communications.

-Nomenclature "large" and "strong" eruptions: Suggest using just one of the terms in the paper (see Figure 1 versus Figure 2).

We indeed forgot to change that nomenclature on Figure 1 (which is now Figure 2) which has been corrected. We checked throughout the manuscript.

Figure 3: Why are you not showing the large eruption figure in the Supplementary Information as well?

We produced that figure specifically for the moderate eruption to highlight differences in the concentration of the sulfur cloud and its location with respect to the tropopause height. However, we concede that it is also valuable to see a similar figure for the large eruption, now included as Figure S5.

REVIEWER COMMENTS

Reviewer #2 (Remarks to the Author):

The authors have done a thorough job revising and addressing my comments carefully. The model and experimental set up became much clearer now, well done. I have two remaining comments, which should be addressed before publication:

-Lines 391-393: The reference to the halogen aerosol and SO₂ effect should give credit to original (aerosol chemistry climate model) studies such as Brenna et al., 2020 ACP. Staunton-Sykes et al is under review and not published yet, but it is written and co-authored by the authors of this paper here.

- Discussion: How do the authors expect the results to be different if the volcanic plume model would be interactive with the atmosphere wrt to radiation, wind, chemistry, microphysics? This needs to be shortly addressed in the ms and not only in the reply to the reviewers, which stated: "For a more extensive discussion into the uncertainties and sensitivities of the eruptive column modelling approach used in this work as well as for a comparison with a 3D eruptive column model result, we continue to direct the reader towards previous studies (Aubry et al. 2016, 2019)."

References:

Aubry, T. J., et al.: Impact of global warming on the rise of volcanic plumes and implications for future volcanic aerosol forcing. *Journal of Geophysical Research: Atmospheres*, 121(22), 13-326, 2016.

Aubry, T. J., Cerminara, M., & Jellinek, A. M.: Impacts of Climate Change on Volcanic Stratospheric Injections: Comparison of 1-D and 3-D Plume Model Projections. *Geophysical Research Letters*, 46(17-18), 10609-10618, 2019.

Brenna, H., Kutterolf, S., Mills, M. J., and Krüger, K.: The potential impacts of a sulfur- and halogen-rich supereruption such as Los Chocoyos on the atmosphere and climate, *Atmos. Chem. Phys.*, 20, 6521–6539, <https://doi.org/10.5194/acp-20-6521-2020>, 2020.

Reviewer #3 (Remarks to the Author):

The paper quantifies the sensitivity of volcanic eruption-induced response to the background climate. The contrast between the large and moderate eruptions is very interesting. The revelation of such a contrast and underlying mechanism is only possible when using a multi-layer model structure as adopted here (1D plume model to estimate injection height, and an emission-driven 3D chemistry-climate model).

The presentation is excellent. At times, the long paragraph can be broken apart. I also suggest adding a final schematic to summarize the many key processes (oftentimes competing or offsetting). This will aid the beginning portion of the Discussion section.

Specific Comment:

Line 114. Why is the rate here not 1:10 as indicated in Line 109?

The separation of the impact due to injection height (as probed in a few limited studies before) and aerosol transformation and transport is very insightful.

Line 119 The purpose of showing everything as a function of the (QBO) phase is unclear. I would assume it makes more sense to show the simulated IH as a function of assumed wind or temperature profile?

Fig 2 Caption. Should be the Initial condition in our UKESM simulations.

Line 161. Fig 3c, e. I think.

Why the faster decay of SO₄? Not obvious to me.

Line 229. causing the decrease in S burden e-folding time. Why is it?

Line 233. "In the modal aerosol scheme implemented" is better to start a new paragraph since it now pivots to size distribution and Fig 7. To be consistent with the flow of Fig 6 to 7, the "aerosol effective radius and S burden e-folding time" in Line 224 can be swapped in order.

Line 929. Delete "an".

Line 328. a stronger meridional stratospheric temperature gradient?? Is this contradicting what you just mentioned in Line 326?

Line 382. Xu and Lamarque also presented similar results using NCAR's model. Xu, Y., and J.-F. Lamarque (2018) Isolating the Meteorological Impact of 21st Century GHG Warming on the Removal and Atmospheric Loading of Anthropogenic Fine Particulate Matter Pollution at Global Scale. *Earth's Future*, 6, 428–440.

Line 404. Where does the 40% number come from? If that's based on ref 28, it might not be directly additive to the 15% (is that referring to radiative forcing or T? If it's radiative forcing, that should be from UKESM1_atmo, right? Or are you referring to the enhancement due to co-injection?)

Line 421. " across all eruption types". Because of this limitation, it's worth adding "tropical" to the title.

Line 441. The paper keeps omitting the BDC changes due to future changes in tropospheric aerosols. I would argue that's as important as stratospheric aerosols, if not more.
<https://agupubs.onlinelibrary.wiley.com/doi/10.1002/2014GL062823>

Reviewer comments are in italic and blue; our responses are in black. All changes to our original submission have been tracked using Microsoft Word track changes tool, except for reference numbers and section.

Reviewer 2

Summary:

The authors have done a thorough job revising and addressing my comments carefully. The model and experimental set up became much clearer now, well done. I have two remaining comments, which should be addressed before publication:

We thank the reviewer again for their comments which helped us greatly improve the manuscript. The two remaining comments have been addressed below.

Lines 391-393: The reference to the halogen aerosol and SO₂ effect should give credit to original (aerosol chemistry climate model) studies such as Brenna et al., 2020 ACP. Staunton-Sykes et al is under review and not published yet, but it is written and co-authored by the authors of this paper here.

We have added the reference suggested (Brenna et al. 2020) and updated the Staunton-Sykes et al. reference as this paper is now accepted.

Discussion: How do the authors expect the results to be different if the volcanic plume model would be interactive with the atmosphere wrt to radiation, wind, chemistry, microphysics? This needs to be shortly addressed in the ms and not only in the reply to the reviewers, which stated: "For a more extensive discussion into the uncertainties and sensitivities of the eruptive column modelling approach used in this work as well as for a comparison with a 3D eruptive column model result, we continue to direct the reader towards previous studies (Aubry et al. 2016, 2019)."

We have now included a more extensive discussion of how a fully interactive setup (i.e. with the eruptive column model coupled to UKESM) may affect our results in the Methods section, at the end of the subsection entitled "Simulation of volcanic eruptions with our combined plume-aerosol-climate modelling framework":

"Our simplified approach mostly has two limitations. First, it neglects the variability of atmospheric conditions at sub-daily timescales during the eruption. Hourly atmospheric profiles were not outputted in our UKESM simulations, but using hourly profiles from the ERA5 reanalysis⁷⁴ at the location of Mount Pinatubo and running the eruptive column model employed for all July 1st profiles in the last 20 years, we find that over 85% of the variability in simulated plume height is associated with interannual variability (as opposed to hourly variability). We conclude that our design enables us to sample well the atmospheric conditions of each climate state. Second, our approach neglects the potential impacts of the early plume development on atmospheric conditions, which could in turn modulate the height reached by the eruptive column. The few studies that have quantified the local, instantaneous response of atmospheric conditions to volcanic eruptions suggest a warming temperature response near the plume top region for plumes composed mostly of SO₂, but a cooling temperature response for those composed mostly of ash⁷⁵. Accounting for such effects would thus require ash to be co-emitted with SO₂ which is not currently possible with UKESM. However, atmospheric conditions

would likely be affected below the spreading plume, downwind of the vent, and it thus remains unclear whether such effects would be of critical importance for modelling the column rise accurately.”

Reviewer 3

The paper quantifies the sensitivity of volcanic eruption-induced response to the background climate. The contrast between the large and moderate eruptions is very interesting. The revelation of such a contrast and underlying mechanism is only possible when using a mufti-layer model structure as adopted here (1D plume model to estimate injection height, and an emission-driven 3D chemistry-climate model).

The presentation is excellent. At times, the long paragraph can be broken apart. I also suggest adding a final schematic to summarize the many key processes (oftentimes competing or offsetting). This will aid the beginning portion of the Discussion section.

We thank the reviewer for their enthusiastic comment and for suggestions that helped us improve the manuscript. Most specific comments have been incorporated in the manuscript and where necessary, we clarify choices made in the responses below. The suggestion of a final schematic is excellent and we have now included a new Figure 9 following your idea. Previous Figure 8 showing coagulation rate has been moved to Supporting Information.

Line 114. Why is the rate here not 1:10 as indicated in Line 109?

Eruption intensity was chosen to reach specific plume heights, which we now explain more explicitly in the Experimental Design subsection in Methods:

“Eruption intensities were chosen to obtain injection heights of ca. 16 and 21 km a.s.l. under HIST atmospheric conditions (Supplementary Figure 2), and they differ by slightly more than a factor of 10 between the two eruption cases.”

The separation of the impact due to injection height (as probed in a few limited studies before) and aerosol transformation and transport is very insightful.

Thank you for this supporting comment.

Line 119 The purpose of showing everything as a function of the (QBO) phase is unclear. I would assume it makes more sense to show the simulated IH as a function of assumed wind or temperature profile?

From a plume height dynamics perspective, showing the height as a function of the vertical-averaged wind speed and stratification would make more sense. However, in terms of initial conditions in UKESM, injection height and QBO phase are the two most important conditions. The aim of figure 2 is to highlight that we explored various combinations of these two initial conditions as well as how the two initial conditions vary between the two climate scenarios chosen. We have thus left this figure unchanged. Note that Supplementary Figures 1-2 clearly show changes in atmospheric conditions and in the eruption intensity-plume height relationship.

Fig 2 Caption. Should be the Initial condition in our UKESM simulations.

Thank you, we have made this change.

Line 161. Fig 3c, e. I think.

Thank you, we have corrected this mistake.

Why the faster decay of SO₄? Not obvious to me.

We are unsure which specific sentence or figure this comment refers to, and whether the reviewer refers to SO₂ or H₂SO₄. Lines 202-205, we explain that the faster SO₂ decay is likely driven by higher OH concentrations in the SSP585 scenario. We extensively discuss the faster H₂SO₄ decay which is driven by the acceleration of the Brewer-Dobson circulation (also see response to comment below)

Line 229. causing the decrease in S burden e-folding time. Why is it?

We have clarified the link between the transport to mid-high latitudes and aerosol loss:

“The sulfuric acid aerosols spread faster to higher latitudes in the SSP585_HIH scenario compared to the historical scenario (Figure 6.d), which in turn decreases the S burden e-folding time as sulfate aerosols sediment from the stratosphere into the troposphere predominantly at mid-high latitudes^{4,5}.”

Line 233. “In the modal aerosol scheme implemented” is better to start a new paragraph since it now pivots to size distribution and Fig 7. To be consistent with the flow of Fig 6 to 7, the “aerosol effective radius and S burden e-folding time” in Line 224 can be swapped in order.

We agree and have implemented both changes suggested.

Line 929. Delete “an”.

Deleted.

Line 328. a stronger meridional stratospheric temperature gradient?? Is this contradicting what you just mentioned in Line 326?

Even though the forcing and stratospheric temperature anomalies are smaller in the HIST scenario, the slower transport to high latitudes result to a higher meridional temperature gradient. We have clarified this in the manuscript:

“Our results are consistent with this picture as the HIST scenario is characterized by a slower transport of the aerosol cloud to high-latitudes (Figure 6.a-f) resulting in a stronger meridional stratospheric temperature gradient as aerosols reside in the tropics for a longer period of time (Figure 9.d-f).”

Line 382. Xu and Lamarque also presented similar results using NCAR’s model.

Xu, Y., and J.-F. Lamarque (2018) Isolating the Meteorological Impact of 21st Century GHG Warming on the Removal and Atmospheric Loading of Anthropogenic Fine Particulate Matter Pollution at Global Scale. Earth’s Future, 6, 428–440.

This is indeed a relevant reference and we have incorporated it lines 384-388:

“Such a feedback mechanism is accounted for in our simulations, but we find a minor decrease (as opposed to the increase in [29], or in the Community Earth System Model⁶³) in tropospheric aerosol optical depth at 550nm between our two chosen climate states (-2% from HIST to SSP585, Table 1).”

Line 404. Where does the 40% number come from? If that’s based on ref 28, it might not be directly additive to the 15% (is that referring to radiative forcing or T? If it’s radiative forcing, that should be from UKESM1_atmo, right? Or are you referring to the enhancement due to co-injection?)

Yes, the 40% number comes from reference 28. We have clarified which % changes refer to radiative forcing and which refer to the temperature response:

“Taken together, the combined enhancement of the surface temperature response driven by an increase in radiative forcing (+15%, this study, estimated using FaIR) and an increase in the surface temperature response to the forcing related to ocean feedbacks (+40%²⁸) could lead to a 60% greater surface cooling for large-magnitude tropical eruptions.”

We agree that the effect of these feedback may not be additive but we are only providing a simple back-of-the-envelope calculation here. Fully coupled (ocean-atmosphere) experiments will be required to determine the combined effect, as mentioned at the end of this discussion.

Line 421. “ across all eruption types”. Because of this limitation, it’s worth adding “tropical” to the title.

We agree and have edited the title accordingly, but the title is now 16 words long and is thus subject to the approval of the editor and publisher.

Line 441. The paper keeps omitting the BDC changes due to future changes in tropospheric aerosols. I would argue that’s as important as stratospheric aerosols, if not more.

<https://agupubs.onlinelibrary.wiley.com/doi/10.1002/2014GL062823>

We thank the reviewer for bringing this to our attention and have mentioned BDC forcing by tropospheric aerosols using the suggested reference:

“This provides a new perspective on how the efficiency of stratospheric aerosol geoengineering may be modulated by changes in atmospheric circulation driven by anthropogenic greenhouse gases emissions⁴⁷, tropospheric aerosols⁷² or stratospheric aerosol geoengineering itself³³.”

REVIEWERS' COMMENTS

Reviewer #2 (Remarks to the Author):

The authors did a good job addressing my points and I have no further comments.

Reviewer #3 (Remarks to the Author):

The supplement contains comments for internal use

Reviewer comments are in italic and blue; our responses are in black.

Reviewer 3

The supplement contains comments for internal use.

Thanks, we have removed these comments.